# Variations in vegetation evapotranspiration affect water yield in high-altitude areas

Yinying Jiao[1,2,3], Guofeng Zhu[1,2,3*], Dongdong Qiu[1,2,3], Siyu Lu[1,2,3], Gaojia Meng[1,2,3], Rui Li[1,2,3], Qinqin Wang[1,2,3], Longhu Chen[1,2,3], Wentong Li[1,2,3]

[1]College of Geography and Environment Science, Northwest Normal University, Lanzhou 730070, Gansu, China
[2]Shiyang River Ecological Environment Observation Station, Northwest Normal University, Lanzhou 730070, Gansu, China
[3]Key Laboratory of Resource Environment and Sustainable Development of Oasis, Gansu Province, Lanzhou 730070, Gansu, China

*Correspondence to*: Guofeng Zhu (zhugf@nwnu.edu.cn)

**Abstract.** Global mountains and plateaus are the main water-producing areas on land. However, under the influence of climate change, the distribution of vegetation and the way water is utilized in these areas have undergone significant changes. As such, understanding the effects of evapotranspiration from high-altitude vegetation on precipitation and runoff is vital in addressing the uncertainties and challenges posed by climate change and anthropogenic transformation. The stable isotopes in water bodies play a crucial role in determining the evapotranspiration capacity of ecosystems and the mechanisms of

precipitation formation. Between 2018 and 2022, we conducted research in the northeastern Qinghai-Tibet Plateau, collected and analyzed stable isotope water data from precipitation, soil water, and Picea crassifolia xylem water to quantify the impact of vegetation transpiration and recirculated water vapor on precipitation. Our findings indicate that transpiration from vegetation accounts for the largest share of evapotranspiration within the entire forest ecosystem, averaging 57%. Therefore, vegetation transpiration is the decisive factor in determining the water yield of inland high-altitude areas. Local

evapotranspiration contributes an average of 28% to precipitation, further enhancing the replenishment of precipitation in high-altitude areas. The warming of global temperatures and human activities are likely to induce shifts in the distribution areas and evapotranspiration regimes of alpine vegetation, potentially altering water resource patterns in the basin. It is necessary to actively adapt to the changes in water resources in the inland river basin.

## 1 Introduction

Projected future scenarios suggest that drought events will become more frequent, severe, and prolonged due to the effects of climate change. This phenomenon is expected to manifest most rapidly and intensely in arid and semi-arid regions (Ault et al., 2020). Large-scale forest ecosystems play a pivotal role in influencing climate through biophysical feedback mechanisms and in altering the global water cycle. The stable hydrogen and oxygen isotopes found in precipitation, plant water, and soil water can effectively trace evaporation within water cycle. During water evaporation, isotopic fractionation occurs as

molecules of differing mass redistribute between the vapor and liquid phases, leaving heavier isotopes such as D and $\delta^{18}O$

predominantly in the liquid phase(Dansgaard et al., 1964). Plant transpiration further enriches heavy isotopes in leaf water, while the heavier and lighter molecules released through stomata remain in equilibrium with the xylem water supply. This mechanism underpins the use of stable isotopes to estimate vegetation transpiration capacity (Farquhar et al., 2007). The Picea crassifolia ecosystem, providing a range of ecological, climatic, and social benefits to the northeastern Tibetan Plateau, exhibits high susceptibility to drought and temperature extremes. Furthermore, climate-related drivers significantly heighten the vulnerability of Picea crassifolia to drought and heat stress, with an anticipated increase in disturbances to its ecosystem as climate change progresses.

Variations in evaporation loss are known to precipitate disturbances in the precipitation and surface water budget (Li et al., 2023). Furthermore, forest evapotranspiration (ET) significantly influences atmospheric moisture convergence and precipitation dynamics, with its impact critically dependent on ambient moisture conditions (Makarieva et al., 2023). Additionally, across different spatial scales, vegetation transpiration capacity and local water availability demonstrate complex and variable relationships. This is specifically reflected in the following aspects. Enhanced evapotranspiration can contribute up to 45% of available water in both local and downwind regions, though this proportion may substantially vary in water-scarce environments (Cui et al., 2022). As precipitation recycling and moisture convergence intensify, the sensitivity of precipitation to evapotranspiration becomes increasingly pronounced (Cheng et al., 2024). In regions characterized by high vegetation cover and elevated topography, moisture recycling significantly enhances regional water resource availability (An et al., 2025). The upward movement of water vapor in the atmosphere, upon merging with advection water vapor, condenses to form precipitation. Consequently, in high-altitude areas with greater vegetation coverage and favorable conditions for moisture convergence, precipitation is typically more abundant. The trajectory and intensity of westerly winds critically determine moisture content and precipitation distribution across Central Asian mountain ranges, particularly during the cold season under the substantial influence of the subtropical westerly jet (Mehmood et al., 2022). Even minor redistributions of atmospheric water can trigger significant cascading effects, inducing substantial shifts in latent heat flux, atmospheric circulation, water transport mechanisms, and precipitation patterns (Hao et al., 2023). Moreover, for high-altitude regions influenced by the cryosphere, the mechanism by which vegetation-dominated water vapor contributes to precipitation and runoff formation remains a critical knowledge gap.

As a vascular plant species, Picea crassifolia plays a crucial role in channeling energy and materials from the environment into terrestrial ecosystems. Its growth, survival, and reproduction significantly influence the ecological functions and structures of other species, both within their habitats and in broader ecological contexts. A significant interaction exists between the vegetation, its drought resilience, and the microclimatic conditions within forests and their ecosystems. This interaction is vital for understanding ecosystem dynamics (Eisenhauer et al., 2021). In this study, we conducted monthly observations and analyses of the xylem water potential, soil water potential, stable isotopes of precipitation, and soil water content of Picea crassifolia in the northeastern Qinghai-Tibet Plateau from April to October for the years 2018 and 2022. These data were utilized to address the following research objectives: (1) To quantify the contribution rates of soil evaporation and vegetation transpiration to the total evapotranspiration of ecosystems; (2) To determine the ratio of

recirculated water vapor in precipitation; and (3) To investigate the evapotranspiration process and its impact on productivity and hydrological convergence in the forest belt of the mountainous region. This study provides a robust foundation for the management of local water resources and the protection of ecological integrity.

## 2 Study area

The Qilian Mountains are located in the central part of the Eurasian continent, on the northeastern edge of the Qinghai-Tibet
Plateau(Figure 1). The eastern region is dominated by water erosion, with large variations in mountainous terrain and an average elevation of over 4,000 meters. Permafrost is developed at elevations of 3,500 to 3,700 meters, and areas above 4,500 meters are characterized by modern glacier development. The region has a plateau continental climate, with hot summers and cold winters, strong solar radiation, and large temperature differences between day and night. The average annual temperature is below 4℃, with extreme highs of 37.6℃ and extreme lows of -35.8℃. The annual sunshine hours
range from 2,500 to 3,300 hours, with a total solar radiation of 5,916 to 15,000 megajoules per square meter. The average annual precipitation is 400 millimeters, and the annual evaporation ranges from 1,137 to 2,581 millimeters. The average wind speed is around 2 meters per second, and the frost-free period lasts from 23.6 to 193 days. The Shiyang River originates from the Daxueshan on the northern side of the Lenglong Ridge in the eastern section of the Qilian Mountains, serving as a major water source for the city of Wuwei. The soil types in the eastern section are diverse, but with low organic
matter content. The distribution of vegetation shows distinct zonal characteristics, with mountainous forest-grassland zones (2,600 to 3,400 meters), subalpine shrub-meadow zones (3,200 to 3,500 meters), and high mountain sub-ice-snow sparse vegetation zones (>3,500 meters) at elevations above 2,700 meters. The main types of natural forest vegetation include Picea crassifolia, Qilian juniper forest, and Chinese pine forest, with Picea crassifolia being the dominant tree species (Zhu et al., 2022).

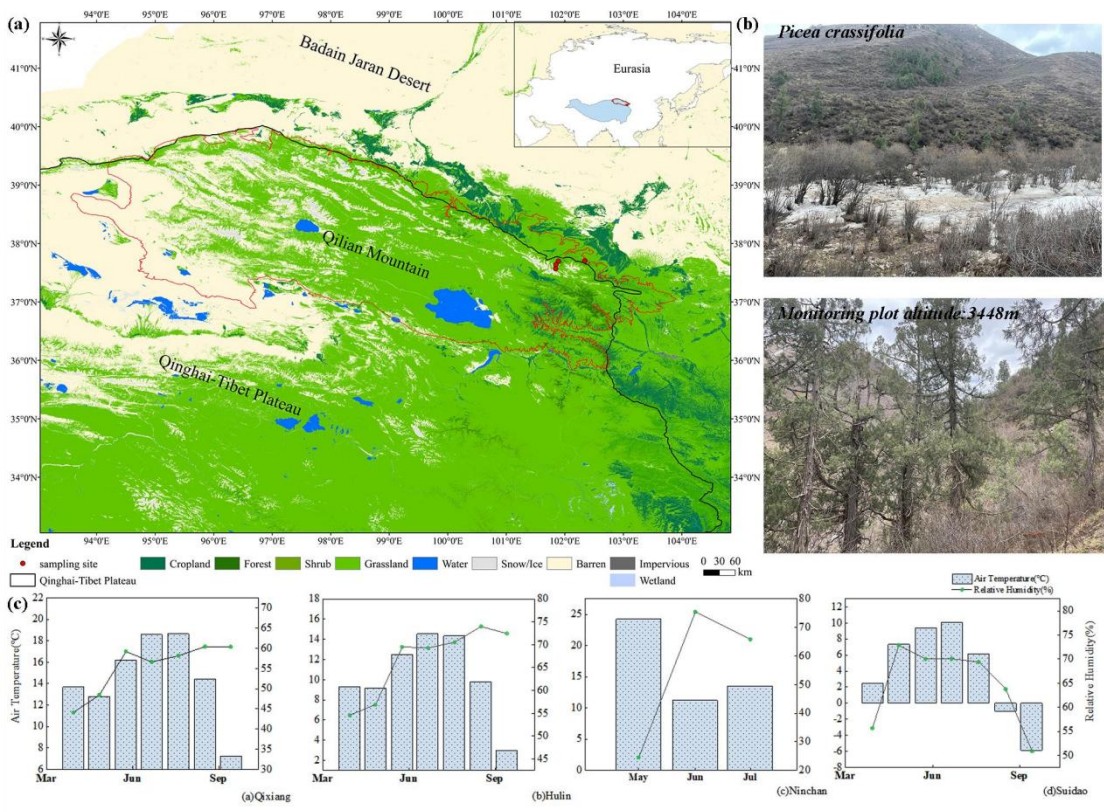

**Figure 1: Overview of the study area. (a) Geographical location of the study area, (b) Growth status of Qinghai spruce, and (c) Seasonal variation of meteorological conditions.**

## 3 Materials and methods

### 3.1 Materials Sources

In this study, we established a stable isotope observation network at four elevation zones (2543m, 2721m, 3068m, 3448m) in terms of vertical height (Table 1). Meteorological data were recorded using an automatic weather station in the years 2018 and 2022. The collected precipitation was subsequently transferred to 100ml containers following each rainfall event. Soil samples were extracted from the sample plot at various depths, specifically at intervals of 0-5 cm, 5-10 cm, 10-20 cm, 20-30 cm, 30-40 cm, 40-50 cm, 50-60 cm, 60-70 cm, 70-80 cm, 80-90 cm, and 90-100 cm, utilizing a soil drill. These samples

were bifurcated, with one portion being stored in a 50 ml glass bottle. This bottle was hermetically sealed with a parafilm and transported to the observation station, where it was marked with the sampling date and subjected to cryopreservation within 10 hours for the purpose of stable isotope analysis. The remaining portion of the soil sample was placed in a 50ml aluminum box to ascertain soil moisture content through a drying method. For the collection of plant samples, scissors were

employed to harvest the xylem stems of vegetation. The bark was removed, and the samples were placed in 50ml glass bottles, which were then sealed and frozen for subsequent experimental analysis.

**Table 1:Sampling location, the meteorological background, and sampling quantity information during the growing season.**

| Parameter | Station | Qixiang | Hulin | Ninchan | Suidao |
|---|---|---|---|---|---|
| | Altitude(m) | 2543 | 2721 | 3068 | 3448 |
| Local climate | Temperature(°C) | 3 | 3.2 | 3.3 | -0.9 |
| | Precipitation(mm) | 262 | 370 | 394 | 475 |
| | Relative humidity(%) | 52.9 | 56.1 | 66.6 | 69.2 |
| Samplin gs number | Precipitation | 53 | 108 | 91 | 135 |
| | Soil water | 220 | 560 | 560 | 560 |
| | Xylem water | 236 | 56 | 56 | 56 |

Surface evapotranspiration data at an elevation of 2700 m were sourced from the MODIS-based daily surface evapotranspiration dataset for the Qilian Mountains. Evapotranspiration data at 3200 m were obtained from high-quality literature that includes field-observed ET values within the elevation range of 2500 to 3400 m to reinforce our research findings. Precipitation data at different elevations were all sourced from the National Tibetan Plateau Data Center.

**3.2 Experimental Analysis**

The isotopic data used in this study mainly include stable isotopes of precipitation, soil water, and xylem water. All isotopic samples were analyzed at the Stable Isotope Laboratory of Northwest Normal University. The precipitation samples were analyzed for hydrogen and oxygen stable isotopes using a liquid water isotope analyzer (DLT-100, Los Gatos Research, USA). After thawing the soil and vegetation samples, they were extracted using a low-temperature vacuum condensation device (LI-2100, LICA United Technology Limited, China), and the extracted water was subjected to isotopic analysis. Each water sample was tested six times to ensure accuracy, with the first two tests considered as interference and only the results of the subsequent four tests were averaged (Zhu et al., 2022). The isotopic measurements are represented by $\delta$, which represents the deviation in parts per thousand of the ratio of two stable isotopes in the sample relative to the ratio in a standard sample. The International Atomic Energy Agency (IAEA) defined the Vienna Standard Mean Ocean Water (VSMOW) in 1968 as the standard for isotopic composition, which is derived from distilled seawater and has a similar isotopic composition to Standard Mean Ocean Water (SMOW).

$$\delta = (\frac{\delta_{Sampling}}{\delta_{Standard}} - 1) \times 1000\textperthousand , \tag{1}$$

## 3.3 Research methods

First, determining the isotopic composition of water vapor formed from precipitation, vegetation, and soil evaporation serves as the foundation for applying different models. Based on the isotopic values of different water vapor sources, the contribution of vegetation's evapotranspiration to the overall ecosystem evaporation can be established, which is a step in identifying the key factors affecting precipitation. Next, an end-member mixing model is used to quantify the contribution ratio of recycled water vapor in precipitation. The results of this analysis will be used to assess the impact of these key factors on the formation of precipitation. The parameters involved in the methods are all listed in Table S1 of the supporting information.

### 3.3.1 Isotopic composition of atmospheric water vapour

The stable isotope composition of moisture in ambient air ($\delta_a$) is calculated as follows(Gibson and Reid, 2014;Skrzypek et al., 2015):

$$\delta_a = \frac{\delta_{rain} - k\varepsilon^+}{1 + k\alpha^+ \times 10^{-3}}, \tag{2}$$

where k=1, or by fitting k to some fraction of 1 as the best fit to the local evaporation line, $\varepsilon^+$ is the isotopic fractionation factor. Defined by $\varepsilon^+ = (\alpha^+ - 1) \times 1000$. $\alpha^+$ about $^2H$ and $^{18}O$ are calculated as follows(Horita and Wesolowski, 1994):

$$10^3 \ln^2\alpha^+ = 1158.8T^3/10^9 - 1620.1T^2/10^6 + 794.84T/10^3 - 161.04 + 2.9992 \times 10^9/T^3, \tag{3}$$

$$10^3 \ln^{18}\alpha^+ = -7.685 + 6.7123 \times 10^3/T - 1.6664 \times 10^6/T^2 + 0.35041 \times 10^9/T^3, \tag{4}$$

Here, $\alpha^+$ is the equilibrium fractionation factor dependent on temperature, and T is the temperature (K). The value of $\alpha^+$ is usually around 1.01, and the value of $\varepsilon^+$ is typically around 10.

### 3.3.2 Isotopic composition of soil evaporation

The Craig-Gordon model was used to calculate the stable isotopic composition of soil evaporation water vapour, $\delta_E$, using the following equation(Yepez et al., 2005).

$$\delta_E = \frac{\alpha_e^{-1}\delta_s - h \quad \delta_a - \varepsilon_{eq} - (1-h)\varepsilon_k}{(1-h) + 10^{-3}(1-h)\varepsilon_k}, \tag{5}$$

where $\alpha_e(>1)$ is the equilibrium factor calculated as a function of water surface temperature, $\delta_S$ is the stable isotopic composition of liquid water at the evaporating surface of the soil (0 ~ 10 cm average stable isotopic composition of soil water), $\delta_a$ is the stable isotopic composition of atmospheric water vapour near the surface, $\varepsilon_{eq}$ represents the equilibrium fractionation corresponding to $\varepsilon_{eq} = (1 - 1/\alpha_e) \times 1000$, $\varepsilon_k$ is the kinetic fractionation factor of oxygen is approximately 18.9‰

and h is the atmospheric relative humidity(Gibson and Reid, 2010). For $\delta^{18}O$, $\alpha_e$ is calculated as follows(Raz-Yaseef et al., 2010):

$$\alpha_e = \frac{1.137 \times 10^6/T^2 - 0.4156 \times 10^3/T - 2.0667}{1000} + 1 ,$$  (6)

Where T is the soil Kelvin temperature (K) at a depth of 5 cm.

### 3.3.3 Isotopic composition of plant transpiration

When transpiration is strong, leaf water is in "isotopic stable state", that is, the isotopic composition of leaf transpiration water is equivalent to that of water absorbed by the roots of rain plants at noon. Therefore, the stable isotopic composition of water in plant xylem can be used to represent the stable isotopic composition of water vapor in plant transpiration. The expression is as follows(Aron et al., 2020):

$$\delta_T = \delta_X ,$$  (7)

where $\delta_X$ is the isotopic ratio of xylem water and $\delta_T$ is the isotopic ratio of transpiration.

### 3.3.4 Evapotranspiration isotope assessment

The Keeling Plot model describes the linear relationship between the oxygen isotope composition of atmospheric water vapour and its reciprocal concentration . The intercept of the curve on the Y-axis represents the oxygen isotopic composition of evapotranspiration ($\delta_{ET}$) and is expressed as(Keeling, 1958;Wang et al., 2015):

$$\delta_a = \frac{C_b(\delta_b - \delta_{ET})}{C_a} + \delta_{ET} ,$$  (8)

Where $\delta_a$ and $C_a$ represent the atmospheric water vapour oxygen isotopic composition (‰) and water vapour concentration in the ecosystem boundary layer, $\delta_b$ and $C_b$ represent the background atmospheric water vapour oxygen isotopic composition and background atmospheric water vapour concentration, and $\delta_{ET}$ is the ecosystem evapotranspiration oxygen isotopic composition.

### 3.3.5 Proportion of vegetation transpiration

The determination of evapotranspiration by means of biotic and abiotic isotopic composition can be used to improve the understanding of community structure and ecosystem function in Picea crassifolia in the northeastern of Tibetan Plateau. Based on the isotope mass balance approach to consider the distribution of major and minor isotopes, the partitioning of evapotranspiration can be achieved using two end-member mixing models (E and T) with the following expression(Kool et al.,2014;Wei et al., 2018):

$$\frac{T}{ET} = \frac{\delta_{ET} - \delta_E}{\delta_T - \delta_E} ,$$  (9)

where $\delta_{ET}$, $\delta_E$ and $\delta_T$ are the isotopic compositions of evapotranspiration (ET), soil evapotranspiration (E) and plant evapotranspiration (T), respectively, and the isotopic values of the three can be obtained by both direct observation and model estimation.

### 3.3.6 Bayesian mixing model

Assuming that precipitation vapor is a mixture of advective water vapour and recirculating water vapour, it is understood that the proportion of both precipitation and precipitation water vapour has the same nature. The proportion of precipitation occupied by recycled vapour is calculated as follow(Kong et al., 2013; Wang et al., 2022):

$$f_{re} = \frac{P_{tr}+P_{ev}}{P_{tr}+P_{ev}+P_{adv}} = f_{tr} + f_{ev} , \tag{10}$$

Where $P_{tr}$, $P_{ev}$ and $P_{adv}$ are precipitation produced by transpiration, surface evaporation and advection, respectively. The relationships among these three types of water vapor and precipitation are as follows (Brubaker et al., 1993; Sang et al., 2023):

$$\delta_{pv} = \delta_{tr}f_{tr} + \delta_{ev}f_{ev} + \delta_{adv}f_{adv} , \tag{11}$$

$$f_{ev} + f_{tr} + f_{adv} = 1 , \tag{12}$$

Where $f_{tr}$, $f_{ev}$ and $f_{adv}$ are the proportional contributions of transpiration, surface evaporation and advection to precipitation, respectively, The $f$ values of three kinds of water vapor were obtained by ISOSource software (https://www.epa.gov/) (Phillips & Gregg, 2001). $\delta_{pv}$, $\delta_{tr}$, $\delta_{ev}$ and $\delta_{adv}$ represent the stable isotopic compositions of precipitation vapor, vegetation transpiration vapor, water-surface evaporation vapor, and advected vapor, respectively. $\delta_{pv}$ is calculated using the following formula:

$$\delta_{pv} = \frac{\delta_p - k\varepsilon^+}{1+k\varepsilon^+} , \tag{13}$$

Where $\delta_p$ represents the stable isotopic composition of precipitable liquid water.

Based on the isotopic relationships among different water phases in either open or closed isotope systems, we use the isotope evaporation model proposed by Craig and Gordon (1965) to determine the stable isotopic composition of soil evaporation vapor ($\delta_{ev}$). The equation is as follows:

$$\delta_{ev} = \frac{\delta_s/\alpha^+ - h\delta_{adv} - \varepsilon}{1-h+\varepsilon_k} , \tag{14}$$

Including the $\delta_S$ is the isotopic composition of liquid water evaporation front, $\delta_{adv}$ is advective vapor, h is relative humidity, $\alpha^+$ is equilibrium fractionation factor, $\varepsilon_k$ is kinetic fractionation factor, $\varepsilon$ is total fractionation factor.

$$\varepsilon = \varepsilon^+/\alpha^+ + \varepsilon_k , \tag{15}$$

$$\varepsilon_k = (1-h)\theta_n C_k , \tag{16}$$

h is the relative humidity, $C_k$ is the kinetic fractionation constant, $\delta^2H$ is 25.1‰, $\delta^{18}O$ is 28.5‰.The weight coefficient $\theta$ of small water body is 1, and $\theta$ of large water body is 0.5. n ranges from 0.5 (fully turbulent transport, with reduced kinetic

fractionation, suitable for lake or saturated soil conditions) to 1 (fully diffused transport, suitable for very dry soil conditions), with a kinetic fractionation coefficient of about 12.2-24.5‰ for $\varepsilon_k$ ($^2$H) in a dry atmosphere (h=0). The kinetic separation coefficient of $\varepsilon_k$ ($^{18}$O) is about 13.8-27.7‰.

The advection water vapor isotope $\delta_{adv}$ in the three-component mixing model needs to be determined by the water vapor isotopic composition at the upwind position. The HYSPLIT model (http://www.arl.noaa.gov/ready/HYSPLIT.html), designed for atmospheric transport analysis using gridded meteorological data, was applied to track moisture sources and analyze air mass trajectories to sampling locations (Stein et al., 2015). We found that the northeastern of Tibetan Plateau was controlled by westerly winds, southeast monsoon and plateau monsoon in June, July and August, and by prevailing westerly winds in September and October. The clustering analysis of air masses in different months shows that air masses accumulate at the northern foot of Qilian Mountains and move from low altitude to high altitude along the valley. Xiying, at 2097 m above sea level, is therefore used as a upwind station from April to October(Zhang et al., 2021). As the air mass ascends along the elevation gradient from this station, the isotope of advective vapor is progressively depleted. Notably, although recycled vapor produced by evapotranspiration does enter the air mass to some extent, most of it escapes to other regions without contributing to precipitation. Consequently, its influence on advective vapor downwind can be considered negligible (Li & Zhang, 2003; Wang et al., 2016). Because this transport process is irreversible and departs from isotopic mass balance in the atmosphere, isotopic fractionation is assumed to be due to Rayleigh distillation(Peng et al., 2011), which is formulated as follows:

$$\delta_{adv} = \delta_{pv-adv} + (\alpha^+ - 1)\ln F , \qquad (17)$$

Where $\delta_{pv\text{-}adv}$ denotes the isotopic composition of precipitable water vapor at the upwind station, which is obtained from Equation (13). The parameter F primarily reflects atmospheric moisture conditions during regional precipitation formation and is commonly represented by the ratio of final to initial water vapor. Since water vapor content is positively correlated with the surface vapor pressure of the whole study area (c=1.657e, where c is the water vapor content in mm, e is the surface vapor pressure in hPa, $R^2$=0.94) (Hu et al., 2015), the surface vapor pressure of each site was used to calculate the value of F.

## 4 Results and analysis

## 4.1 Hydrogen and oxygen isotope variations in different water bodies

During the growth season of Picea crassifolia, precipitation stable isotopes display distinct fluctuation patterns(Table 2). In the initial growth phase, the hydrogen and oxygen isotope ratios exhibit relatively low values. With the progressive rise in temperature, the rate of water evaporation and subsequent loss escalates, resulting in an isotopic enrichment. The mean $\delta^2$H value in precipitation throughout the growth season is recorded at -45.52‰, with fluctuations ranging from -151.88‰ to 63.43‰. Similarly, the mean $\delta^{18}$O value stands at -7.75‰, exhibiting fluctuations between -31.49‰ and 14.79‰. The isotopic composition in the wood tissues does not show significant depletion or enrichment, displaying a fluctuation range from -76.95‰ to 23.87‰ for $\delta^2$H and from -11.92‰ to 24.77‰ for $\delta^{18}$O. Compared to precipitation and wood tissues,

shallow soil water demonstrates a lesser enrichment of heavy isotopes, with a reduced fluctuation extent observed during the late spring and early summer period.

**Table 2:Stable isotopes of different water bodies during the growing season.**

| Average / Period | δ²H/‰ | | | δ¹⁸O/‰ | | |
|---|---|---|---|---|---|---|
| | Precipitation | Xylem water | Soil water (0~10cm) | Precipitation | Xylem water | Soil water (0~10cm) |
| April | -69.15 | -39.02 | -53.10 | -10.25 | 2.56 | -7.10 |
| May | -39.09 | -29.78 | -45.38 | -7.61 | 4.44 | -6.42 |
| June | -31.29 | -45.83 | -46.08 | -5.74 | -2.83 | -6.12 |
| July | -32.39 | -47.63 | -47.71 | -5.33 | -0.97 | -7.06 |
| August | -48.88 | -44.55 | -68.85 | -7.79 | -2.06 | -9.07 |
| September | -29.38 | -42.62 | -49.20 | -6.46 | -1.83 | -6.79 |
| October | -68.43 | -44.57 | -54.88 | -11.06 | -2.25 | -7.96 |

Using the Global Meteoric Water Line (GMWL) as a reference standard, regional water lines are influenced by factors like moisture source and re-evaporation during precipitation. The intersection points of the Local Meteoric Water Line (LMWL), Soil Water Line (SWL), and Local Evaporation Line (LEL) can reveal recharge relationships between different water bodies. Their slopes reflect key information including local temperature and humidity characteristics, and the degree of evaporative fractionation in water bodies. Variations in the Local Meteoric Water Line (LMWL) across different vertical gradients are primarily influenced by temperature and humidity. Notably, relative humidity remains consistently low at all four measured elevations within the forest, causing the LMWL to be lower than the Global Meteoric Water Line (GMWL). At an elevation of 2543 meters, which marks the lowest tree growth layer, temperatures can reach up to 20℃ in July, with the LMWL showing a slope of 6.74. At 2721 meters, the average temperature during the growing season is 10.4℃, peaking at 16.45℃ in July and an average relative humidity of 64.38%. The slope of the LMWL at this elevation is 7.02(Figure 2d). At the Suidao station, located at an elevation of 3448 meters, the slope of the precipitation regression line is 7.75. This value is close to the GMWL's slope but exhibits the largest deviation from the local evaporation line, as depicted in (Figure 2a). In the forest's lower layer, the Soil Water Line (SWL) is narrower and closer to the local evaporation line, indicating more pronounced evaporative fractionation and dynamic fractionation compared to the other three sampling zones. The SWL slopes are less steep than those of the LMWL, indicating that precipitation is the primary source of soil moisture replenishment.

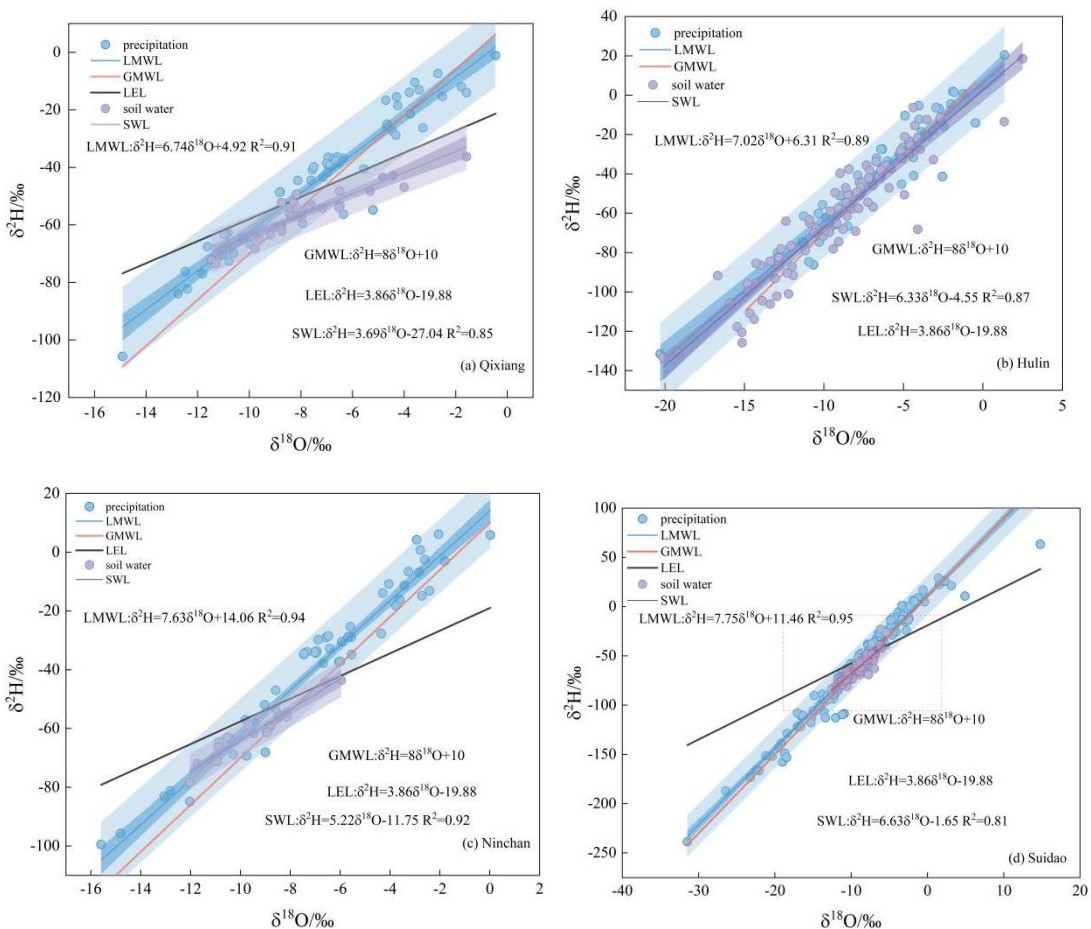

**Figure 2:(a) Qixiang, (b) Hulin, (c) Ninchan, (d) Suidao. Compare the distributions and fittings of precipitation and soil water stable isotopes at the four locations.**

The d-excess parameter measures how much precipitation deviates from the Global Meteoric Water Line, reflecting the impact of re-evaporation on isotope fractionation. Higher d-excess values indicate stronger non-equilibrium evaporation

during regional moisture transport. During the precipitation process, unsaturated water vapor leads to non-equilibrium fractionation, as indicated by an average d-excess value of 16.58‰ throughout the growing season (Figure 3a). Lower relative humidity in May and September resulted in higher d-excess values compared to other months, indicating more pronounced non-equilibrium evaporation during precipitation events. From June to August, the fluctuations in deuterium values are gradual; however, significant variations begin from mid-August onwards. This trend suggests that local

evaporation intensifies over time, influenced by temperature and relative humidity, resulting in increased rates of non-equilibrium evaporation. At higher elevations, the soil moisture content across all soil layers remains above 30%, influenced by rainfall and snowmelt (Figure 3b). By the end of the growing season, decreasing temperatures lead to leaf fall, resulting in

the formation of a litter layer on the forest floor. This layer plays a pivotal role in retaining soil moisture, underscoring the dynamic interactions between vegetation, soil, and atmospheric conditions throughout the season.

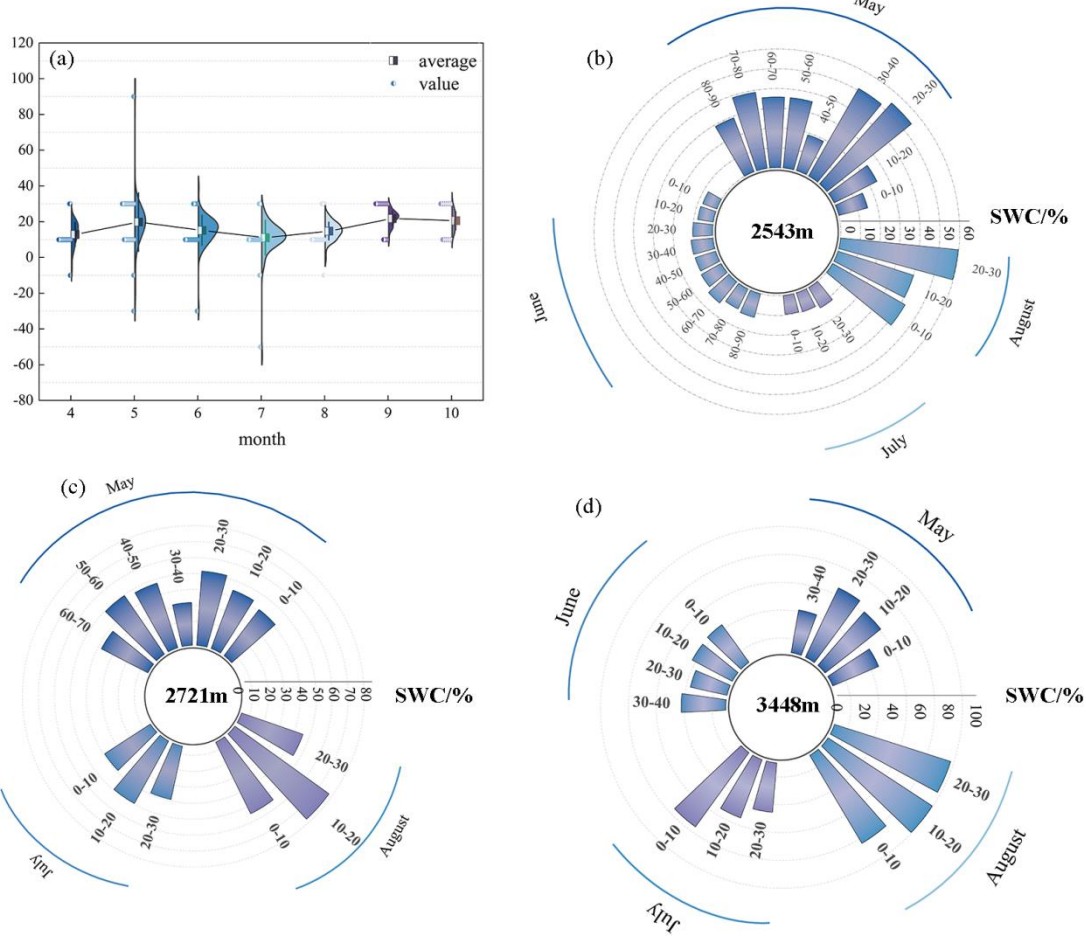

**Figure 3: (a) The average variation of d-excess during the growing season within the gradient of 2543 to 3448 m; (b), (c), and (d) represent the soil moisture content at different depths at elevations of 2543 m, 2721 m, and 3448 m, respectively.**

## 4.2 Soil evaporation, plant transpiration and ecosystem evapotranspiration

**Table 3:The isotopic composition of vegetation transpiration($\delta_T$), soil evaporation($\delta_E$), and ecosystem evapotranspiration($\delta_{ET}$) at
different elevations during the growing season(* represents missing value).**

| Site | Type | April | May | June | July | August | September | October |
|------|------|-------|-----|------|------|--------|-----------|---------|
| Qixiang | $\delta_T$ | 2.22 | -5.87 | -4.59 | -0.72 | -1.72 | -1.78 | -2.26 |
| | $\delta_E$ | -30.32 | -28.68 | -27.33 | -29.12 | -28.68 | -26.32 | -27.27 |
| | $\delta_{ET}$ | -20.19 | -20.05 | -11.63 | -9.87 | -13.56 | -15.85 | -21.56 |
| Hulin | $\delta_T$ | -5.34 | -3.58 | -4.13 | -0.34 | -2.35 | -4.25 | -1.97 |
| | $\delta_E$ | -29.68 | -27.28 | -25.8 | -27.75 | -24.56 | -25.21 | -27.88 |
| | $\delta_{ET}$ | -21.59 | -22.36 | -8.93 | -10.17 | -11.57 | -18.8 | * |

| | | | | | | | | |
|---|---|---|---|---|---|---|---|---|
| Ninchan | $\delta_T$ | * | -3.45 | -1.98 | -1.05 | -6.68 | * | * |
| | $\delta_E$ | * | -20.57 | -26.31 | -29.08 | -18.22 | -18.15 | -18.22 |
| | $\delta_{ET}$ | * | * | -12.46 | -7.57 | * | * | * |
| | $\delta_T$ | * | -8.45 | -6.98 | -6.05 | -6.68 | * | * |
| Suidao | $\delta_E$ | -29.79 | -27.32 | -27.91 | -23.83 | -28.78 | -25.8 | -28.06 |
| | $\delta_{ET}$ | -24.31 | -16.14 | -15.19 | -10.07 | -18.05 | -23.02 | -18.65 |

The Keeling plot method was used to analyze the stable isotope composition of ecosystem evapotranspiration (Figure 4). Its principle involves linearly fitting the water vapor concentration in the ecosystem boundary layer against the oxygen isotope composition, with the intercept on the y-axis representing the stable isotope value of $\delta_{ET}$. The results indicate that at different heights within the distribution of deciduous trees, the average $\delta_{ET}$ value is -22.59‰. Throughout the entire growing season, $\delta_{ET}$ does not consistently decrease with increasing elevation. Specifically, near the treeline, there are higher stable isotope values, but in the middle and upper layers of the forest, there is a minimal value, indicating lower and less stable isotopic fractionation in that layer. At an elevation of 3448m, as the number of deciduous trees decreases and shrubs become dominant, the $\delta_{ET}$ value is -21.81‰(Table 3). We found that the stable isotope $\delta_E$ of soil evaporation at depths of 0-10cm is more enriched at lower elevations, particularly in April and May when the isotopic enrichment is more pronounced. From June to August, due to a significant increase in vegetation coverage, soil evaporation intensity decreases. In the early stage of the growing season, when leaves have not fully developed, the stable isotope composition of the xylem exhibits a relatively depleted characteristic. In July and August, when leaves are fully expanded, temperatures rise, and the rainy season in mountainous areas commences, transpiration becomes more intense.

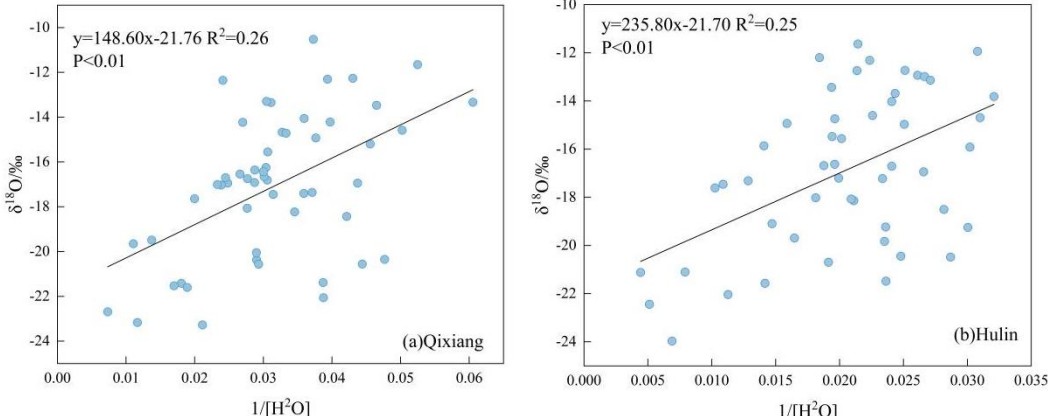

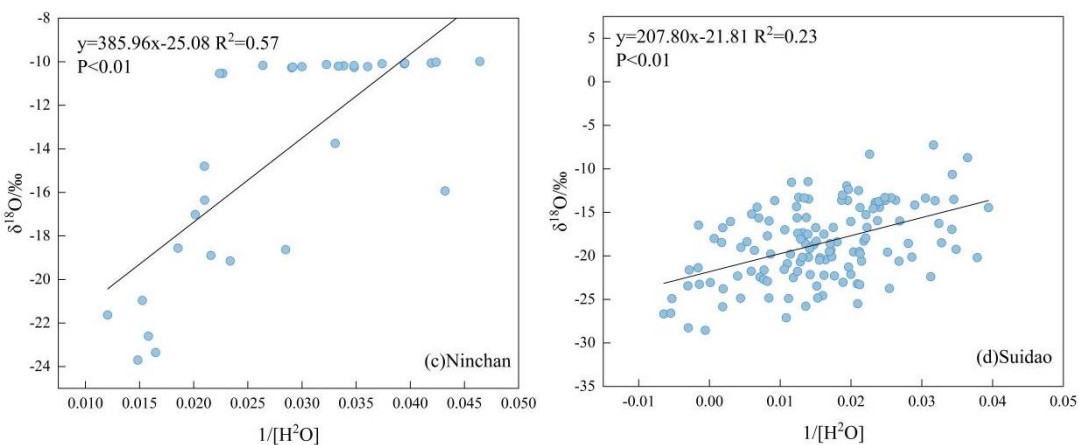

**Figure 4:Each sampling point is fitted with a trend line based on the Keeling plot method.**

**4.3 T/ET assessment of Picea crassifolia ecosystem in different months**

We found that the canopy closure of deciduous trees significantly influences the evapotranspiration of the entire ecosystem (Figure 5). In April and May, as temperatures rise, surface vegetation exhibits weaker growth, resulting in a higher proportion of soil evaporation within the ecosystem, while transpiration by vegetation remains relatively low. During the rainy season in June to August, vegetation experiences vigorous growth, and transpiration reaches its peak in July. In September and October, soil evaporation becomes more dominant as temperatures, relative humidity, and rainfall gradually decrease, and deciduous tree leaves become wilted. At lower elevations, the T/ET ratio fluctuates between 0.20 and 0.70 in a different pattern, while above the treeline, transpiration ratios fluctuate between 0.20 and 0.80 in a similar pattern. Overall, summer is characterized as the peak season for transpiration, with a minimal contribution from soil evaporation.

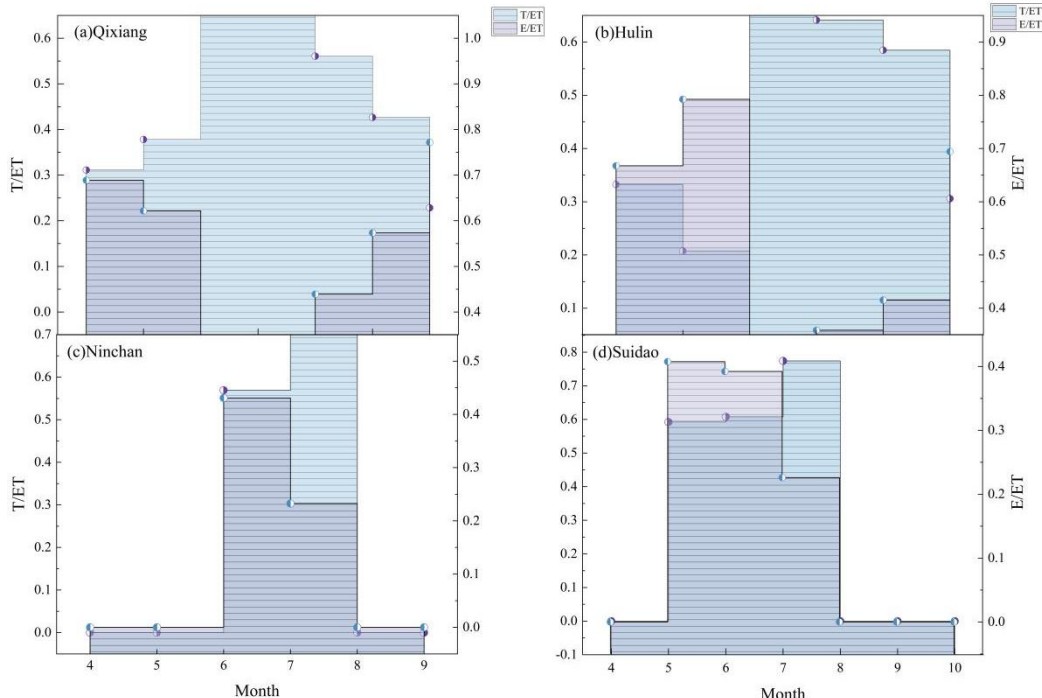

**Figure 5:(a), (b), (c), and (d) represent the proportion of soil evaporation and vegetation transpiration in the evapotranspiration of the ecosystem at different sampling points (0 represents a missing value).**

## 5 Discussions

### 5.1 Hydrological effects of changes in evapotranspiration

The analysis results indicate that the proportion of vegetation in ecosystem evapotranspiration and recycled water vapor is significantly greater than that of the soil. Furthermore, the enhanced evapotranspiration capacity accelerates the cycling rate of water vapor, promoting precipitation formation. Within the altitude gradient of 2700-3000 m, the contribution of advected water vapor gradually declines (Table S1), while the contribution ratio of recycled water vapor to precipitation gradually increases (Figure 7). This is attributed to relatively high temperatures in the lower layers of the forest and the dense presence of Picea crassifolia in the middle to upper layers, resulting in a higher transpiration ratio ($f_{tr}$) throughout the entire growing season. In July, which typically experiences higher temperatures and increased vegetation activity, the ratio of vegetation transpiration($f_{tr}$) is significantly higher compared to other months. Both the early and late stages of the growing season exhibit noticeably higher evaporation ratios ($f_{ev}$) compared to other months, with the middle and upper parts of the forest having a greater proportion of evaporated vapor. The average advected vapor ratio ($f_{adv}$) is 72%, with contributions exceeding 70% for all months except June and July. In mountainous areas, recycled water vapor contributes an average of 28% to precipitation, indicating that the increase in evapotranspiration promotes the occurrence of local precipitation events.

This also means that cross-regional water vapor transport and local-scale recycled water vapor alleviate regional water resource crises and enhance the ecosystem's resilience to drought by regulating the redistribution of precipitation and surface water (Quan et al., 2024).

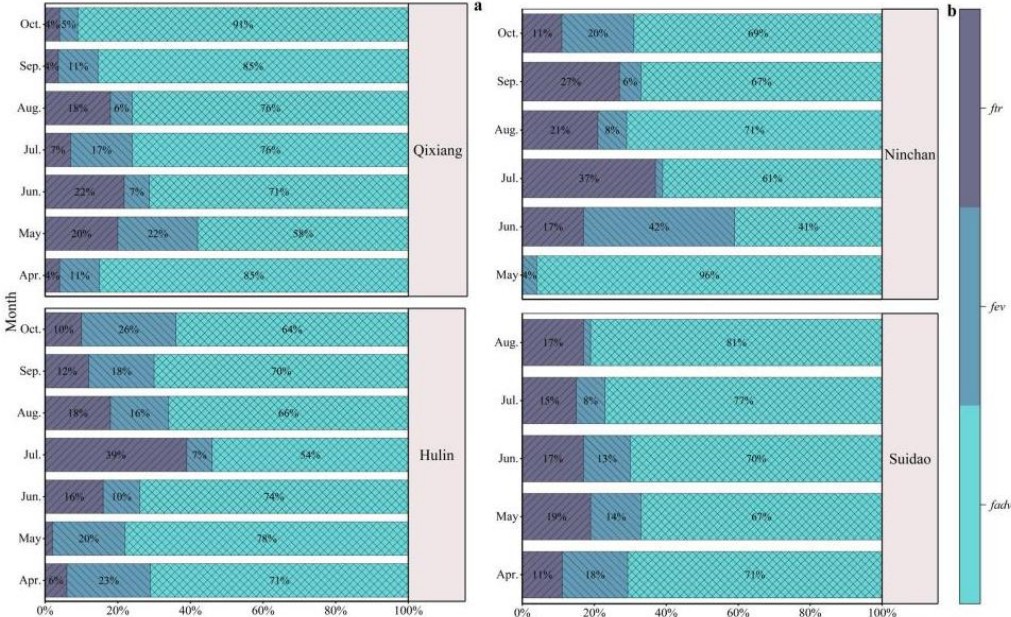

**Figure 7:Comparison of $f_{adv}$ (advective water vapour contribution), $f_{ev}$ (surface evaporation water vapour contribution) and $f_{tr}$ (plant transpiration water vapour contribution) for each of period.**

### 5.1.2 Impact on surface runoff

The formation of runoff is regulated by elevation. In the distribution area of Picea crassifolia, the level of evapotranspiration is much higher than precipitation, making it difficult for surface runoff to form. Comparing the differences of monthly potential evapotranspiration, surface evapotranspiration and precipitation in Picea crassifolia (Table 4), the results clearly showed that rainfall fluctuated between 0-16mm, and the maximum rainfall was 15.7 mm in April, while the minimum value of surface evapotranspiration is 41.8 mm and the minimum value of potential evapotranspiration is 44.1mm. The difference between $ET_P$ and ET shows that there is no effective water accumulation in all months. Between 2500 m and 3400 m, Picea crassifolia-dominated forests cover approximately 38.5% of the watershed area. However, such vegetation exerts only a minor influence on total annual runoff (He et al., 2012). With rising temperatures, precipitation becomes the primary water source for these forests, whereas during dry and cold seasons, snowmelt and seasonal permafrost are integral for both forest water supply and baseflow formation. Seasonal permafrost above 3000 m shapes baseflow from November through April. Over the growing season (2700 – 3400 m), rainfall is the governing factor behind runoff dynamics (Liu et al., 2023), and snowmelt accounts for 33% of the water absorbed by Picea crassifolia (Zhu et al., 2022). Overall, forests at mid-elevation largely rely on rainfall, secondarily on meltwater and seasonal permafrost, while their ability to intercept rainwater for runoff

generation is relatively limited. Notably, the thickness of seasonal permafrost in this region has declined by 7.4 cm per decade (Qin et al., 2016), further restricting its influence on runoff. From this, an important conclusion can be drawn: at elevations between 2500 m and 3400 m, local evapotranspiration substantially exceeds precipitation. Regional runoff is predominantly sustained by precipitation, with forest interception and seasonal permafrost having limited effects on runoff. The feeble runoff generation potential in this region also indicates that afforestation would significantly increase water loss via ET, posing a threat to water distribution and utilisation.

**Table 4:Variations in evapotranspiration(ET) and precipitation(P) across different elevation zones during april to october.**

| Altitide/m | Parameters | April | May | June | July | August | September | October | Data Sources |
|---|---|---|---|---|---|---|---|---|---|
| 2700 | ET/mm | 51.5 | 66.3 | 93.3 | 108.9 | 110.7 | 81.2 | 41.8 | (Yao et al., 2017) |
| 2700 | P/mm | 29 | 31.2 | 103.5 | 90.3 | 66.9 | 39.3 | 10.4 | (Zhao et al., 2019;2020) |
| 3200 | ET/mm | 37.5 | 90 | 95 | 107.5 | 107.5 | 65 | 65 | (Yang et al., 2019) |
| 3200 | P/mm | 27.9 | 43.9 | 62.3 | 39.4 | 56.6 | 42.7 | 23.6 | (Zhao et al., 2021;2022) |

Some studies suggested that reducing forest density will result in less ET in seasonally dry forests. That reduced ET can be converted into increased groundwater and runoff to supply downstream social water (Wyatt, O'Donnell, & Springer, 2015). It has also been claimed that in some cases, the transient increase in water availability through reduced forest density can actually contribute to subsequent increases in vegetation cover and ultimately reduce runoff (Tague et al., 2019). By assessing the hydrological effects of afforestation through the water cycle in the Asia-Pacific region, it was found that in 7 of the 15 water-deficient areas, positive effects such as increased yield, precipitation, soil moisture and reduced drought risk were achieved through afforestation, and it was confirmed that the water-water cycle had a strong impact and evapotranspiration was increased (Teo et al., 2022). The water vapour content produced by forest transpiration is much higher than that lost by soil surface evaporation, most of the precipitation is intercepted and infiltrated by surface vegetation, and part of the soil water involved in infiltration is absorbed by the root zone of vegetation. Because of plants' high interception and evaporation ability and the absorption of groundwater by root zone, the proportion of transpiration was significantly higher than that of evaporation(Su et al., 2014). In this study zone, upwardly transported advective water vapor progressively diminishes with increasing altitude. The vegetation at elevations from 2500 to 3200 meters supplies abundant evapotranspiration water vapor into the water cycle, the acceleration of which boosts local precipitation(Figure 6). Within the elevation range of 2543 to 3448 m, precipitation and seasonal permafrost are the main sources of groundwater recharge.

Over the past 55 years, the seasonal permafrost has decreased at a rate of about 7.4 cm per decade (Li et al., 2016). Correlation analysis shows that when seasonal permafrost decreases, soil moisture in the upper layer increases (Qin et al., 2016). As vegetation transpiration increasingly consumes precipitation, soil water, and groundwater, the remaining groundwater becomes more limited. Consequently, within this elevation range, the contribution of groundwater to runoff formation is minimal.

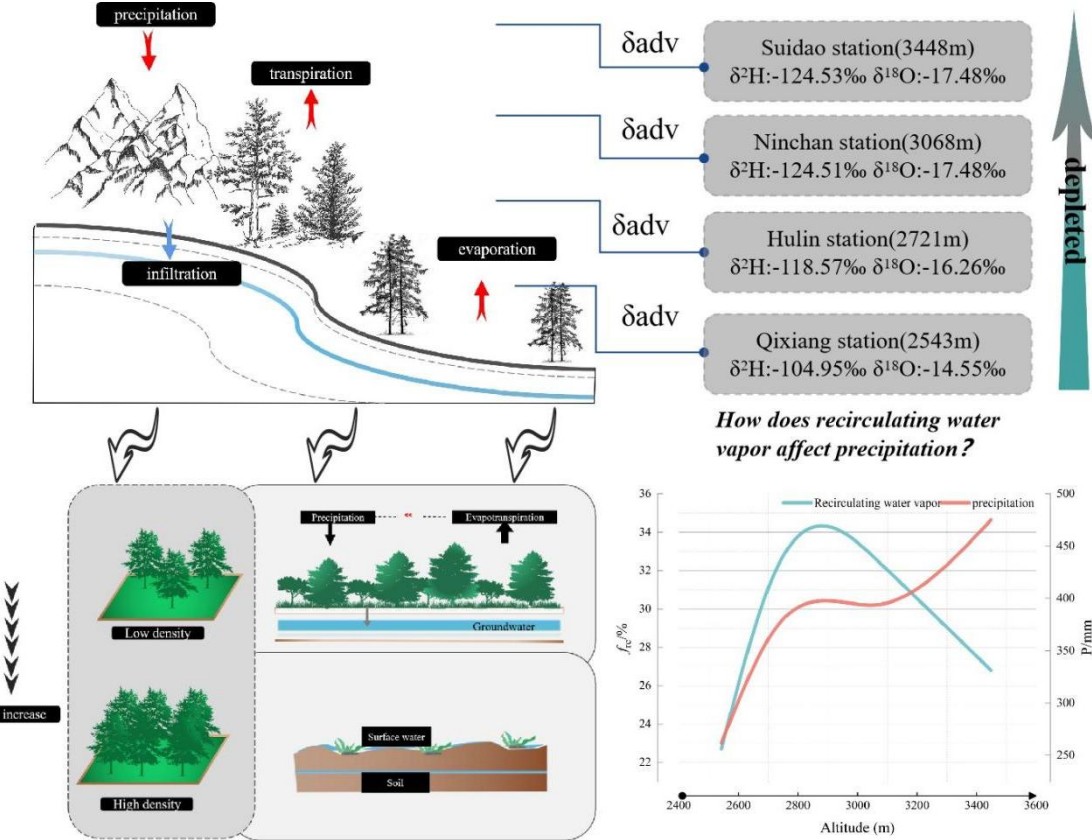

Figure 6:Conceptual model of the hydrological effects of changes in evapotranspiration.

## 5.2 Uncertainty analysis

A higher sample size can reduce the margin of error. Therefore, we utilized isotopic data from four sites over a two-year period to evaluate the model. We used 404 xylem samples to calculate the contribution ratio of transpiration to ecosystem evapotranspiration. We examined the uncertainty of the model evaluation. When analyzing the evaporation characteristics in a semi-arid natural environment using the Craig-Gordon isotopic model, we first eliminated the influence of solar radiation and other meteorological variables on the calculation results. We focused on temperature, relative humidity, water vapor, and the initial isotopic values of water bodies. Particularly in semi-arid environments, the variations in temperature and relative

humidity are crucial (Hernández-Pérez et al., 2020). To verify the calculation results, we found a strong correlation between the isotopes of soil evaporation and relative humidity, as demonstrated by the fitting of $\delta_E$ against relative humidity and temperature(Figure 8). This also indicates the reliability of the results obtained through the Craig-Gordon isotopic model. We employed the Keeling plot method to calculate $\delta_{ET}$, which is based on isotopic mass balance and a two-endmember mixing model. This method assumes that the isotopic composition of the background atmosphere and source remains constant, with a very low probability of isotopic spatial variation (Good et al., 2012; Kool et al., 2014). Due to the higher reliability of oxygen isotopes compared to hydrogen isotopes (Han et al., 2022; Kale et al., 2022), we solely used oxygen isotopes to calculate the T/ET values. The results indicate that transpiration significantly outweighs evaporation during July and August, which aligns with previous research findings (Zhu et al., 2022). The correlation between T/ET and soil moisture content suggests that soil moisture is a crucial factor driving the variations in transpiration and evaporation ratios. Additionally, the estimation of isotopic composition of advected water vapor from the upwind sites contributes to increased uncertainty. In our study area, the sites are predominantly influenced by valley winds, with water vapor moving from the valley bottom to higher altitudes. Therefore, we selected lower elevation areas in the valley bottom as the source region for advected water vapor (Zhang et al., 2021).

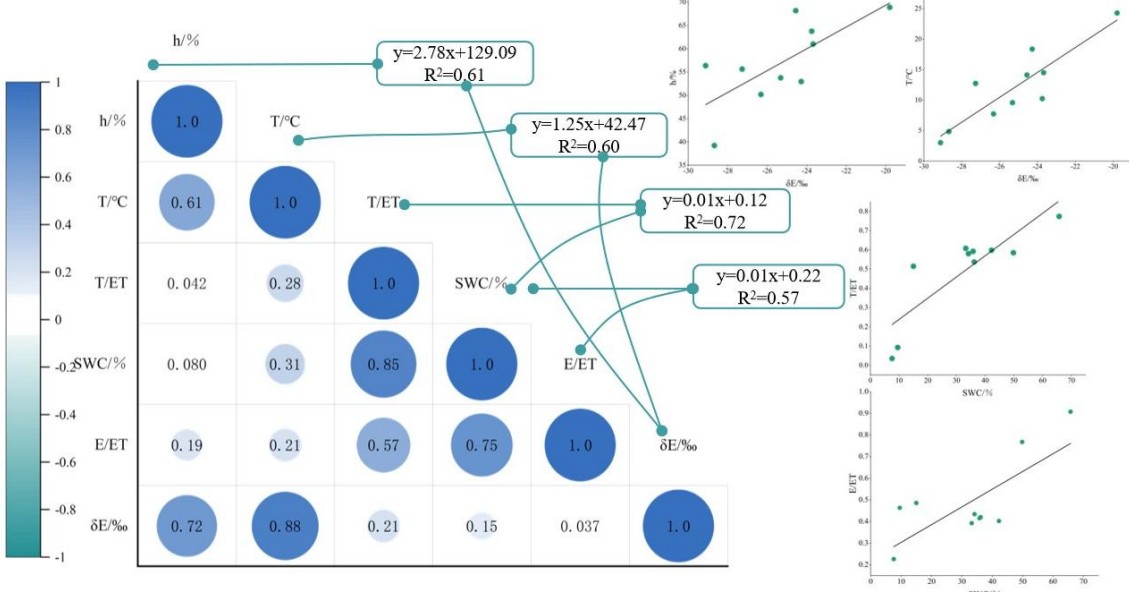

**Figure 8:Correlation analysis of factors affecting uncertainty in impact assessment.**

## 6 Conclusions

This study leverages isotopic data from field observations (2018-2022) and model simulations to investigate the dynamics of evapotranspiration in the northeastern Tibetan Plateau, aiming to elucidate its relationship with local water cycling and

hydrological impacts. The findings reveal that evaporation and transpiration rates peak during July and August, indicating that transpiration from Picea crassifolia plays a more dominant role than soil evaporation in these periods. Quantitative analysis of plant transpiration and soil evaporation contributions to total evapotranspiration yielded an average T/ET ratio of 0.57 over the study period, reaching a maximum of 0.77 in July. Consequently, it is evident that transpiration by forest trees is the primary component of evapotranspiration within the Picea crassifolia ecosystem. Further examination of the hydrological effects associated with Picea crassifolia evapotranspiration demonstrates that monthly evapotranspiration volumes are at least threefold higher than precipitation, significantly limiting the potential for surface runoff formation in this region. Comparative analysis of atmospheric water vapor contributions from precipitation across spring, summer, and autumn reveals that the June to August period marks the peak transpiration season for Picea crassifolia, contributing up to 25% of total atmospheric water vapor, whereas surface evaporation accounts for only 18%. Within the 2543 to 3448m elevation range, the average value of $f_{re}$ is 28%, it is indicated that the water vapor cycle generated by vegetation evapotranspiration has increased the total precipitation in high-altitude mountain areas. In light of global warming, drought, water scarcity, and climate changes driven by relative humidity alterations have significantly impacted the ecological communities, ecosystem functions, services, and land-climate interactions of Picea crassifolia. It is imperative to recognize the critical role of evapotranspiration in depleting rainfall within this forest belt, underscoring its significance for local water resource management and ecological conservation.

**Data availability**

The data that support the findings of this study are available on request from the corresponding author, stable isotope data are not publicly available due to privacy or ethical restrictions. Precipitation and surface evapotranspiration data are available from the National Tibetan Plateau Scientific Data Centre(TPDC).

**Author contribution**

Yinying Jiao and Guofeng Zhu conceived the idea of the study; Gaojia Meng, Siyu Lu analyzed the data; Dongdong Qiu, Qinqin Wang, Rui Li, Longhu Chen participated in the drawing; Yinying Jiao wrote the paper; Wentong Li checked the format. All authors discussed the results and revised the manuscript.

**Competing interests**

The authors declare that they have no conflict of interest.

## Acknowledgements

This research was financially supported by the National Natural Science Foundation of China (42371040, 41971036), the Key Natural Science Foundation of Gansu Province (23JRRA698), the Key Research and Development Program of Gansu Province (22YF7NA122), the Cultivation Program of Major key projects of Northwest Normal University (NWNU-LKZD-202302), the Oasis Scientific Research achievements Breakthrough Action Plan Project of Northwest Normal University (NWNU-LZKX-202303). In addition, we would like to express our gratitude to the Cold and Arid Research Network (CARN) of Lanzhou University for providing the series of precipitation data that supported some of the research results, these datasets are provided by the National Tibetan Plateau / Third Pole Environment Data Center (http://data.tpdc.ac.cn).

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
