# Peer review of "Variations in vegetation evapotranspiration affect water yield in high-altitude areas"

_EGUsphere, 2024_

## Author Response (AR3)

**Response to Reviewer #1**

We sincerely appreciate Reviewer #1's recognition of the scientific merits of our manuscript and the valuable suggestions provided. We are confident that, following these major revisions, the quality of our manuscript will be substantially improved. Moreover, we have found it exceptionally helpful that, along with detailed and highly professional feedback points, you also offered contextual background, reasoning, and relevant reference materials, this level of thoroughness is truly commendable.

We will address every one of your comments with the utmost care, and we hope that, after our revisions, the manuscript will meet your expectations and justify the time and effort you have generously invested. The red words, sentences, and subsections in the marked-up manuscript represent our editing changes.

**Responses to the major comments**

**Comment one:**

"Evapotranspiration is a key process on land in several aspects. It is a proxy of biological productivity as well as an important process of energy conversion with significant implications for regional and global atmospheric moisture transport and climate. Evapotranspiration is difficult to measure directly. So all data on evapotranspiration are welcome and the authors appear to have performed much work to arrive at their first result. I have several relatively minor comments listed below that could clarify the significance of this result, but otherwise it is meaningful and can be published. I have to note though that I am not an experimentalist and could not follow all the procedures in detail, so perhaps an extra reading by another specialist could be helpful. I would also recommend the authors to explain briefly to the readers the physics and ecology of isotopic fractionation, e.g. following and/or quoting the excellent review by Farquhar et al. 2007 https://doi.org/10.1104/pp.106.093278 "Heavy Water Fractionation during Transpiration""

**Response:** Each of your questions has been essential for enhancing the credibility of our results. To further improve the manuscript's readability and clarity, we have **referenced** the classic publication "Heavy Water Fractionation during Transpiration." Additionally, we have expanded the introduction to include the physical and ecological principles of isotope fractionation in the first paragraph. **The supplementary details** are as follows:

"The stable hydrogen and oxygen isotopes found in precipitation, plant water, and soil water can effectively trace evaporation within hydrological systems. During water evaporation, isotopic fractionation occurs as molecules of differing mass redistribute between the vapor and liquid phases, leaving heavier isotopes such as D and $\delta^{18}O$ predominantly in the liquid phase(Dansgaard et al., 1964). Plant transpiration further enriches heavy isotopes in leaf water, while the heavier and lighter molecules released through stomata remain in equilibrium with the xylem water supply. This mechanism underpins the use of stable isotopes to estimate vegetation transpiration capacity (Farquhar et al., 2007)."

**Comment two:**

"The authors model this change using the so-called Rayleigh distillation (their Eq. 17). This equation assumes that the sole source of changes for the isotopic ratio is rainfall. This assumption allows one to estimate the local isotopic ratio of the advected water from the isotopic ratio upwind and the ratio of upwind and downwind water vapor concentrations. However, this assumption is at best problematic. Because as the air moves downwind, its vapor composition is influenced not only by rainfall, but also by moisture recycling, i.e. by added evaporation over the area along which it is moving. Evaporated moisture has a lower content of heavy isotopes, so moisture recycling along the air pathway changes the isotopic composition of the water vapor. This is especially relevant when evaporation and precipitation are of the same order of magnitude as in the case in the studied locations. A relevant study is that of Natali et al. 2022 https://doi.org/10.1016/j.jhydrol.2022.128497 " Is the deuterium excess in precipitation a reliable tracer of moisture sources and water resources fate in the western Mediterranean? New insights from Apuan Alps (Italy)". It shows that there can be fundamentally different interpretations of changing isotopic ratios.

I would therefore request that the authors should provide a thorough justification for their assumptions behind the calculation of the isotopic composition of the advected moisture. Personally I cannot see a straightforward solution given the many unknowns and big uncertainties."

**Response:** We have carefully considered your concern that upwind plain-derived water vapor may be affected by recycled water vapor (surface evapotranspiration) during its transport. Our reply is divided into two parts. First, we clarify the **relationship** between the findings of Natali et al. and our own study. Second, we offer a more rigorous and detailed discussion of the **reliability** of determining the isotopic composition of plain water vapor by calculating water vapor content ratios at different locations.

**Part 1**

From Natali et al. (2022) in *Journal of Hydrology*, we learn that in the Apuan Alps of Italy, the d-excess in precipitation increases with elevation, seemingly due to sub-cloud evaporation driving the spatial variability of d-excess in mountainous regions. **Their results suggest** that the evolution of d-excess during water vapor transport may compromise the reliability of d-excess as an indicator for source regions and transport pathways of atmospheric water vapor.

The d-excess value reflects how stable hydrogen and oxygen isotopes in water samples deviate from the Global Meteoric Water Line (GMWL); it is typically employed to indicate isotope fractionation during evaporation and to reflect atmospheric temperature and humidity at the time of precipitation formation. Because stable isotopes in precipitation exhibit altitude and temperature effects, d-excess also shows an altitude effect (Dansgaard, 1964). However, research in humid regions indicates that sub-cloud evaporation weakens with increased elevation, resulting in higher d-excess values at higher altitudes, whereas in arid regions, d-excess exhibits an inverse altitude effect, decreasing with increasing elevation (Pang et al., 2011). Some studies use d-excess to quantify the relative contributions of recycled vapor (surface evaporation plus plant transpiration) and plain vapor to precipitation (Zhang et

al., 2021), although Natali et al. questioned the accuracy of this method. **Conversely**, Xia et al. found that in mid-latitude continental regions, the contribution of recycled vapor (i.e., evapotranspiration vapor) during warm and dry seasons had only a **minimal effect** on the d-excess.

When we calculated the isotopic composition of advective water vapor, we **did not** rely on the d-excess parameter. Instead, we used **Equation (13)** to recalculate the isotopic composition of precipitable water vapor at the upwind station. As you have noted, however, the influence of other water vapor sources on upwind water vapor transport must also be taken into account. Through our literature review, we found that Li and Zhang (2003) in Central Asia's arid regions concluded that any recycled vapor entering the air mass in transit **could be neglected**, as its overall contribution to the atmospheric column remains **minimal** and it largely exits the air mass **without contributing** to downwind precipitation. Wang et al. (2016) confirmed these findings.

**References**

Dansgaard W. Stable isotopes in precipitation[J]. tellus, 1964, 16(4): 436-468.

Xia L I, Guang-xin Z. Research on precipitable water and precipitation conversion efficiency around Tianshan Mountain Area[J]. Journal of Desert Research, 2003, 23(5): 509.

Pang Z, Kong Y, Froehlich K, et al. Processes affecting isotopes in precipitation of an arid region[J]. Tellus B: Chemical and Physical Meteorology, 2011, 63(3): 352-359.

Zhang F, Huang T, Man W, et al. Contribution of recycled moisture to precipitation: A modified D-excess-based model[J]. Geophysical Research Letters, 2021, 48(21): e2021GL095909.

Xia Z, Welker J M, Winnick M J. The seasonality of deuterium excess in non-polar precipitation[J]. Global Biogeochemical Cycles, 2022, 36(10): e2021GB007245.

Wang S, Zhang M, Che Y, et al. Contribution of recycled moisture to precipitation in oases of arid central Asia: A stable isotope approach[J]. Water Resources Research, 2016, 52(4): 3246-3257.

**Part 2**

In an effort to reduce the unknowns and uncertainties inherent to our assumptions, we read multiple articles and two seminal books, thereby reinforcing the feasibility of Equation (17) in our manuscript. **The key details** are as follows:

"As the air mass ascends along the elevation gradient from this station, the isotope of advective vapor is progressively depleted. Notably, although recycled vapor produced by evapotranspiration does enter the air mass to some extent, most of it escapes to other regions without contributing to precipitation. Consequently, its influence on advective vapor downwind can be considered negligible (Li & Zhang, 2003; Wang et al., 2016). Because this transport process is irreversible and departs from isotopic mass balance in the atmosphere, isotopic fractionation is assumed to be due to Rayleigh distillation(Peng et al., 2011), which is formulated as follows:

$$\delta_{adv} = \delta_{pv-adv} + (\alpha^+ - 1)\ln F, \tag{17}$$

Where $\delta_{pv\text{-}adv}$ denotes the isotopic composition of precipitable water vapor at the upwind station, which is obtained from Equation (13). The parameter F primarily reflects atmospheric moisture conditions during regional precipitation formation and is commonly represented by the ratio of final to initial water vapor. Since water vapor content is positively correlated with the surface vapor pressure of the whole study area (c=1.657e, where c is the water vapor content in mm, e is the surface vapor pressure in hPa, R2 200 =0.94)(Hu et al., 2015), we used the surface vapor pressure of each site to calculate the value of F."

**References**

Clark, I.D., & Fritz, P. (1997). Environmental Isotopes in Hydrogeology (1st ed.). CRC Press. https://doi.org/10.1201/9781482242911.

Natali S, Doveri M, Giannecchini R, et al. Is the deuterium excess in precipitation a reliable tracer of moisture sources and water resources fate in the western Mediterranean? New insights from Apuan Alps (Italy)[J]. Journal of Hydrology, 2022, 614: 128497.

Hu W, Yao J, He Q, et al. Spatial and temporal variability of water vapor content during 1961–2011 in Tianshan Mountains, China[J]. Journal of Mountain Science, 2015, 12: 571-581.

Xia L I, Guang-xin Z. Research on precipitable water and precipitation conversion efficiency around Tianshan Mountain Area[J]. Journal of Desert Research, 2003, 23(5): 509.

Wang S, Zhang M, Che Y, et al. Contribution of recycled moisture to precipitation in oases of arid central Asia: A stable isotope approach[J]. Water Resources Research, 2016, 52(4): 3246-3257.

**Comment three:**

Regarding the third conclusion, that evaporation is several times higher than rainfall, my question is, how does this forest survive? I suspect that this conclusion could be an artifact of comparing evapotranspiration from a global gridded dataset to locally measure precipitation. Local evapotranspiration has very big uncertainties even on an annual scale. The authors refer to the study of Yao et al. 2022 https://doi.org/10.1016/j.agrformet.2017.04.011 "Improving global terrestrial evapotranspiration estimation using support vector machine by integrating three process-based algorithms". Yao et al.'s Fig. 10, lower right panel, shows that estimated evapotranspiration can considerably exceed (by ~100 mm/year) the actual one derived from water balance measurements in river basins. Arguably local values on a small scale should be even more prone to errors. I would therefore urge the authors to find additional arguments to justify their conclusions which cannot be published as they now stand.

**Response:** To address your concerns and further solidify our findings, we conducted an extensive **literature review**. We must concede that your skepticism regarding certain inferences in **Section 5.1.2** is both reasonable and pivotal. Consequently, we plan to **re-evaluate and revise the conclusions** pertaining to how forest cover affects surface runoff in that section.

**1. Improve the accuracy of data in the table**

Stable water isotopes alone can quantify only rates of evapotranspiration (ET), which is why relying on reanalysis data or integrating established research is essential to elucidate how forests allocate and manage water resources. Nevertheless, as you rightly pointed out, using reanalysis data to derive small-scale ET estimates introduces substantial uncertainties. Our main task is thus to refine the reliability of reanalysis-derived dataset results in the context of existing studies, and to reassess how ET influences runoff.

Due to the large elevation span in mountainous areas and the varying runoff conditions across different vegetation distribution zones, analyzing the differences in evapotranspiration and precipitation at different altitude gradients in the Picea crassifolia distribution area can help enhance the credibility of our results.We selected high-quality literature that includes field-observed ET values within the 2500-3400m elevation range to reinforce our research findings. Additionally, during our data review, we found some problems: some months were missing rainfall information, the data only came from one narrow elevation area (around 2700 m), and this made it hard to understand the full picture of water yield in the region. To solve these issues, we used more comprehensive datasets from the National Tibetan Plateau Data Center (Zhao et al., 2019; 2020; 2021; 2022). These datasets was collected at elevation points of 2700 m and 3200 m within the study area, giving us a much better understanding of the formation of regional runoff and improving the overall reliability of our research findings.

After re-collecting and compiling the information, we have improved the details in the table as follows:

**Table 4:Variations in evapotranspiration(ET) and precipitation(P) across different elevation zones during april to october.**

| Altitide/m | Parameters | April | May | June | July | August | September | October | Data Sources |
|---|---|---|---|---|---|---|---|---|---|
| 2700 | ET/mm | 51.5 | 66.3 | 93.3 | 108.9 | 110.7 | 81.2 | 41.8 | (Yao et al., 2020) |
| 2700 | P/mm | 29 | 31.2 | 103.5 | 90.3 | 66.9 | 39.3 | 10.4 | (Zhao et al., 2019; 2020) |
| 3200 | ET/mm | 37.5 | 90 | 95 | 107.5 | 107.5 | 65 | 65 | (Yang et al., 2019) |
| 3200 | P/mm | 27.9 | 43.9 | 62.3 | 39.4 | 56.6 | 42.7 | 23.6 | (Zhao et al., 2021; 2022) |

**References and Datasets**

Yang, Y., Chen, R., Song, Y., et al.: Sensitivity of potential evapotranspiration to meteorological factors and their elevational gradients in the Qilian Mountains, northwestern China, J. Hydrol., 568, 147–159, 2019.

Yao, Y., Liu, S., and Shang, K.: Daily MODIS-based Land Surface Evapotranspiration Dataset of 2019

in Qilian Mountain Area (ETHi-merge V1.0), National Tibetan Plateau / Third Pole Environment Data Center, doi:10.11888/Meteoro.tpdc.270407, 2020.

Zhao, C., Zhang, R., Zhao, C.: Cold and Arid Research Network of Lanzhou university (an observation system of Meteorological elements gradient of Sidalong Station, 2021), National Tibetan Plateau / Third Pole Environment Data Center, doi:10.11888/Atmos.tpdc.272365, 2022.

Zhao, C., Zhang, R., Zhao, C.: Cold and Arid Research Network of Lanzhou university (an observation system of Meteorological elements gradient of Sidalong Station, 2020), National Tibetan Plateau / Third Pole Environment Data Center, doi:10.11888/Meteoro.tpdc.271378, 2021.

Zhao, C., Zhang, R., Wang, Y.: Cold and Arid Research Network of Lanzhou university (an observation system of Meteorological elements gradient of Dayekou Station, 2019), National Tibetan Plateau / Third Pole Environment Data Center, doi:10.11888/Meteoro.tpdc.270799, 2020.

Zhao, C., Zhang, R., Wang, Y.: Qilian Mountains integrated observatory network: Cold and Arid Research Network of Lanzhou University (an observation system of meteorological elements gradient of Dayekou Station, 2018), National Tibetan Plateau / Third Pole Environment Data Center, doi:10.11888/Geogra.tpdc.270169, 2019.

**2. How does the forest survive under these conditions?**

This is indeed a crucial question. **Section 5.1.2** of our manuscript will expand on the following points:

**-Importance of elevation gradients**

The Qilian Mountains range from roughly 4000 to 5000 m above sea level, whereas our study area lies between 2543 and 3448 m. Strong zonal differences driven by elevation largely determine how forests persist here.

**- Cryosphere elements above 3400 m**

Above 3400 m, seasonal and permanent permafrost and snow dominate the hydrology (Li et al., 2019). Since 1990, climate warming in these high-altitude regions has outpaced warming at mid- and lower-elevation forest belts, prolonging melt periods of glaciers and permafrost. Glacial ice, snow cover, frozen soils, and rainfall together form the primary runoff production and recharge sources for these mountains at high latitudes.

**- Forces shaping forest water use between 2500 m and 3400 m**

Within this elevation, Picea crassifolia-dominated forests cover approximately 38.5% of the watershed area. However, such vegetation exerts only a minor influence on total annual runoff (He et al., 2012). With rising temperatures, precipitation becomes the primary water source for these forests, whereas during dry and cold seasons, snowmelt and seasonal permafrost are integral for both forest water supply and baseflow formation. Seasonal permafrost above 3000 m shapes baseflow from November through April. Over the growing season (2700 – 3400 m), rainfall is the governing factor behind runoff dynamics (Liu et al., 2023), and snowmelt accounts for 33% of the water absorbed by Picea crassifolia (Zhu et al., 2022). Overall, forests at mid-elevation largely rely on rainfall, secondarily on meltwater and seasonal permafrost, while their ability to intercept rainwater for runoff generation is relatively

limited. Notably, the thickness of seasonal permafrost in this region has declined by 7.4 cm per decade (Qin et al., 2016), further restricting its influence on runoff.

From these observations, **we draw an important conclusion**: "at elevations between 2500 m and 3400 m, local evapotranspiration substantially exceeds precipitation. Regional runoff is predominantly sustained by precipitation, with forest interception and seasonal permafrost having limited effects on runoff. The feeble runoff generation potential in this region also indicates that afforestation would significantly increase water loss via ET, posing a threat to water distribution and utilisation."


I would recommend to make a table listing and explaining all the notations used in the article.

**Response:** Indeed, this step is essential. In the supplementary material, we have added a comprehensive table containing all the symbols used in the manuscript, along with their definitions, corresponding equations, and relevant references. The specific details of the table are as follows:

**Table S1: Details of all parameters in the equations**

| Parameter | Formula | Meaning | References |
|---|---|---|---|
| $\delta_a$ | Eq.(2) | Isotopic composition of atmospheric water vapor, ‰ | (Gibson and Reid, 2014;Skrzypek et al., 2015) |
| $\alpha^+$ | Eq.(3) and (4) | Temperature-dependent equilibrium fractionation factor, ‰ | (Horita and Wesolowski, 1994) |
| $\varepsilon^+$ | $\varepsilon^+=(\alpha^+-1)\times1000$ | Equilibrium fractionation factor for liquid-vapor phase transition, ‰ | (Horita and Wesolowski, 1994) |
| $\delta_E,\delta_S$ | Eq. (5) | Isotopic composition of soil evaporation | (Yepez et al., 2005) |

| Symbol | Equation | Description | Reference |
|---|---|---|---|
| | | vapor, ‰ | |
| | | Isotopic composition of shallow soil water, ‰ | |
| $\alpha_e$ | Eq. (6) | Equilibrium coefficient calculated based on water surface temperature, ‰ | (Raz-Yaseef et al., 2010) |
| $\varepsilon_{eq}$ | $\varepsilon_{eq} = (1-1/\alpha_e) \times 1000$ | Equilibrium coefficient calculated from $\alpha_e$, ‰ | (Gibson and Reid, 2010) |
| $\delta_X$ $\delta_T$ | Eq. (7) | Isotopic composition of xylem water, ‰ Isotopic composition of vegetation transpiration vapor, ‰ | (Aron et al., 2020) |
| $C_a$ $\delta_b$ $C_b$ $\delta_{ET}$ | Eq. (8) | Water vapor concentration in the ecosystem boundary layer, g/m³ Isotopic composition of background atmospheric water vapor, ‰ Water vapor concentration in background atmosphere, g/m³ Isotopic composition of ecosystem evapotranspiration, ‰ | (Keeling, 1958;Wang et al., 2015) |
| $f_{re}$ | Eq. (10), ISOSource software (https://www.epa.gov/) | Recycled water vapor ratio in precipitation, % | (Phillips & Gregg, 2001) |
| $\delta_{pv}$ | Eq.(11) and (13) | Isotopic composition of precipitation vapor, ‰ | (Brubaker et al., 1993) |
| $\delta_{ev}$ | Eq. (14) | Isotopic composition of soil evaporation vapor, ‰ | (Brubaker et al., 1993) |
| $\delta_{adv}$ | Eq. (17) | Isotopic composition of advective vapor, ‰ | (Brubaker et al., 1993) |
| $\delta_{tr}$ | $\delta tr = \delta T$ | Isotopic composition of vegetation transpiration vapor, ‰ | (Evaristo et al., 2015) |
| $\varepsilon$ | Eq. (15) | Total fractionation factor, ‰ | (Skrzypek et al., 2015) |
| $\varepsilon_K$ $C_K$ | Eq. (16) | Kinetic fractionation factor, ‰ Kinetic fractionation constant, $\delta^2H$ is 25.1‰, $\delta^{18}O$ is 28.5‰ | (Skrzypek et al., 2015) |
| $F$ | Eq. (17) | Ratio of final to initial water vapor at different sites, dimensionless | (Peng et al., 2011) |

Please also derive an explicit formula for f_re (defined in Eq. 10) using isotopic fractions delta, as it is

your main result.

Response: This was an oversight on our part. We sincerely appreciate your careful observation. We will revise Equation (10) and clarify how the value of fre is obtained. The specific details of these modifications are as follows:

$$f_{\text{re}} = \frac{P_{tr}+P_{ev}}{P_{tr}+P_{ev}+P_{adv}} = f_{tr} + f_{ev} \; , \tag{10}$$

The $f$ values of three kinds of water vapor were obtained by ISOSource software (https://www.epa.gov/)(Phillips & Gregg, 2001).

Please avoid reporting delta to the accuracy of four (!) digits. Apparently the data do not have this accuracy.

Response:Thank you for your thoughtful comments. Are you referring to the four-decimal-place precision for the isotope values? I'm not entirely sure I fully understand your point. We measured all stable isotope values of liquid water using the liquid water isotope analyzer in our laboratory. To minimize the interference from other compounds in the water, we used the LWIA-Spectral Contamination Identifier (SCI) V1.0 software for correction. To eliminate the memory effect of the liquid water isotope analyzer, we discarded the first two measurements of the water samples and used the average of the last four measurements as the final result, ensuring that the isotope values are precise to two decimal places.

**Responses to the minor comments (numbers are line numbers)**

10: The abstract could better reflect the actual content of the study. Currently it does not mention the isotopic analysis at all despite that is the focus. The authors do not really study any climatological change, so referring to that in the abstract appears to be somewhat misleading.

Response:Thank you for the reminder. Including the isotope information in the abstract indeed enhances readers' understanding of the manuscript. We will add the **following statement**:

"The stable isotopes in water bodies play a crucial role in determining the evapotranspiration capacity of ecosystems and the mechanisms of precipitation formation."

10:"posed by climate change" – please consider adding "and anthropogenic transformation", which is a major driver of vegetation changes

Response:Thank you very much for your valuable addition. We will incorporate this statement into the original text.

15: Even if local precipitation makes a 28% contribution to local rainfall, it does not automatically become "the main driver behind the increase in precipitation at high latitudes". Again, changes in

precipitation are not investigated in the paper.

**Response:** We need to **reconsider** whether the term "main driver" is appropriate. After reviewing authoritative literature in the field, we fully agree with your perspective and have revised the sentence to:

"Local evapotranspiration contributes an average of 28% to precipitation, further enhancing the replenishment of precipitation in high-altitude areas."


40: "This phenomenon is more pronounced in high-altitude areas, which are characterized by greater vegetation coverage and favorable conditions for the confluence of water vapor, resulting in more abundant precipitation". Do you mean "in those high-latitude areas where there is a greater vegetation coverage and favorable conditions…" Please check

**Response:** Your statement is more coherent and clearer. The **details of the revision** for the following sentence are as follows:

"In those high-altitude areas with greater vegetation coverage and conditions favorable for moisture convergence, precipitation is typically more abundant."

75: Legend to Figure 1 "changes in meteorological conditions". Do you mean seasonal variation of meteorological conditions? What time period do the data shown in the lower panel refer to?

**Response:** We apologize for the inconvenience caused by unclear label fonts and legends. We have increased the font size of the legend (time period: April to October) and revised the figure caption to:

"Figure 1: Location of the study area and seasonal variation of meteorological conditions."

The details of the revised image are as follows:

[Figure]

120: Equation 2: please define epsilon^+

**Response:** Thank you for your reminder. We have added the following details:

"where k=1, or by fitting k to some fraction of 1 as the best fit to the local evaporation line, $\varepsilon^+$ is the

isotopic fractionation factor. Defined by $\varepsilon^+=(\alpha^+-1)\times1000$."

120: Please check this phrase "Defined by. about $^2H$ and $^{18}O$ are calculated as follows"

**Response:** Due to our previous mistake of choosing a formula format when inserting symbols, some symbols did not display correctly. We will carefully review all formulas in the manuscript to ensure this issue does not recur. The additional details are as follows:

"$\alpha^+$ about $^2H$ and $^{18}O$ are calculated as follows(Horita and Wesolowski, 1994):"

125: Please give a characteristic value for alpha^+. Is it large compared to 1?

**Response:** Based on our calculations, the value of alpha^+ is usually around 1.01, and it often exceeds 1.01. The value of epsilon^+ is typically around 10.

130 and elsewhere: Please pay attention to get all the lower indexes right. Currently many of them are printed not as indexes but as plain text.

**Response:**Thank you for the reminder. We will review all subscript characters throughout the manuscript and ensure they are no longer presented as plain text.

150: What is the difference if any between delta_A and delta_a?

**Response:** Both delta_A and delta_a represent the isotopic composition of atmospheric water vapor and share the same meaning. We will revise the text to consistently use "delta_a"

155: picea should be Picea

**Response:** We sincerely apologize again for any inconvenience caused. We have thoroughly reviewed the entire manuscript and revised it to consistently use "Picea."

165: Equation 10: please revise this equation or the preceding line, currently it is not the proportion of precipitation occupied by advective vapor but the proportion occupied by evaporation.

**Response:** We have revised the line before Equation (10). The specific changes are as follows:

"The proportion of precipitation occupied by recycled vapour is calculated as follow(Kong et al., 2013; Wang et al., 2022):"

170: "This can be calculated" please clarify what is "This"

**Response:** We replaced "This" with a clearer explanation referencing the relationship between water vapor and precipitation. The revised sentence is:

"Where Ptr, Pev and Padv are precipitation produced by transpiration, surface evaporation and advection, respectively. The relationships among these three types of water vapor and precipitation are as follows (Brubaker et al., 1993; Sang et al., 2023):"

175: Please define "C-G"

**Response:** We will add information on the meaning, applicability, and references of the "C-G model" here. The specific revisions are as follows:

"Based on the isotopic relationships among different water phases in either open or closed isotope systems, we use the isotope evaporation model proposed by Craig and Gordon (1965) to determine the stable isotopic composition of soil evaporation vapor($\delta_{ev}$). The equation is as follows:"

175: Please check the phrase "are the stable isotopes in precipitating transpiration, transpiration, surface evaporation and advective vapour, respectively". Should it be "water vapor in the precipitating column (?), transpiration, surface evaporation and advective vapour"?

**Response:** There is indeed a labeling issue here, as $\delta_{pv}$ represents the isotopic composition of precipitation vapor. We have **revised the statement** as follows:

"$\delta_{pv}$, $\delta_{tr}$, $\delta_{ev}$, and $\delta_{adv}$ represent the stable isotopic compositions of precipitation vapor, vegetation transpiration vapor, water-surface evaporation vapor, and advected vapor, respectively."

What is delta_p in Eq. 13, is this for precipitating liquid water? Please explain.

**Response:** In Equation 13, $\delta_p$ refers to the isotopic composition of precipitable liquid water, while $\delta_{pv}$ represents the isotopic composition of precipitable water vapor. To resolve ambiguity, we have added the following explanation to the manuscript:

$$\delta_{pv} = \frac{\delta_p - k\varepsilon^+}{1 + k\varepsilon^+} , \tag{13}$$

"In the equation, $\delta_p$ represents the stable isotopic composition of precipitable liquid water."

180 and elsewhere: "steam" – please use "vapor" consistently everywhere

**Response:** We appreciate your attention to terminology consistency. Ensuring "vapor" is standardized throughout the text strengthens methodological clarity. Let us know if further refinements are needed!

190: Please give more details about the HYSPLIT model

**Response:** We appreciate this suggestion and are happy to provide additional details about the **HYSPLIT model**. We have addressed this oversight by adding the following section to the revised manuscript:

"The HYSPLIT model (http://www.arl.noaa.gov/ready/HYSPLIT.html), designed for atmospheric transport analysis using gridded meteorological data, was applied to track moisture sources and analyze air mass trajectories to sampling locations (Stein et al., 2015)."

195: "as follow" "as follows", "winds tation" "wind station"

**Response:** Thank you for identifying this oversight. The error will be corrected in the revised

manuscript.

200: Since rainfall is positively correlated with the surface vapor pressure of the whole study area (c=1.657e, where c is the water vapor content in mm, e is the surface vapor pressure in hPa, R2=0.94), Do you mean the total water vapor content is correlated with surface vapor pressure? Rainfall has the units of mm per unit time, not mm. So correlation between total water vapor content and surface vapor pressure is not equivalent to correlation between rainfall and surface vapor pressure.

**Response:**You are absolutely correct. The correlation between total water vapor content and surface vapor pressure cannot be equated with that between rainfall and surface vapor pressure. This was an oversight in our original phrasing. The revised text now reads:

"Since water vapor content is positively correlated with the surface vapor pressure of the whole study area (c=1.657e, where c is the water vapor content in mm, e is the surface vapor pressure in hPa, $R^2$=0.94) (Hu et al., 2015), the surface vapor pressure of each site was used to calculate the value of F."

215-240: Please could you clarify in greater details how these figures (with the meteorological water line) are relevant to your main results. What parameters from these figures are you using and where?

**Response:**We will add explanations of these parameters in the manuscript and discuss their impact on our main results. The details are as follows:

"Using the Global Meteoric Water Line (GMWL) as a reference standard, regional water lines are influenced by factors like moisture source and re-evaporation during precipitation. The intersection points of the Local Meteoric Water Line (LMWL), Soil Water Line (SWL), and Local Evaporation Line (LEL) can reveal recharge relationships between different water bodies. Their slopes reflect key information including local temperature and humidity characteristics, and the degree of evaporative fractionation in water bodies."

"The d-excess parameter measures how much precipitation deviates from the Global Meteoric Water Line, reflecting the impact of re-evaporation on isotope fractionation. Higher d-excess values indicate stronger non-equilibrium evaporation during regional moisture transport."

235: please define "lc-access" and explain how it is relevant.

**Response:** Both d-excess and lc-excess characterize how precipitation deviates from the Global Meteoric Water Line, reflecting the impact of re-evaporation on isotope fractionation and enrichment. Since d-excess and lc-excess have similar implications, we will remove the sentences about lc-excess.

**Reference books and literature:**

Lyu, S., Wang, J., Song, X., et al.: The relationship of δD and δ18O in surface soil water and its implications for soil evaporation along grass transects of Tibet, Loess, and Inner Mongolia Plateau, J. Hydrol., 600, 126533, 2021.

Gu, W. (Ed.): Isotope Hydrology, Science Press, Beijing, China, 2011.

265: "in a distinct pattern" do you mean "opposite"?

**Response:** We revised the phrase "in a distinct pattern" to "different" to improve clarity. While "distinct pattern" could imply a stronger contrast (e.g., "opposite"), our data specifically highlights quantitative differences rather than diametrically opposed trends. This adjustment aligns the wording more precisely with the observed variations.

230: In May and September, the evaporation rate of water vapor increases due to relatively higher humidity levels compared to other periods.

Please check this. Should not evaporation decrease when the humidity is higher?

**Response:** We sincerely appreciate your careful review. You are absolutely correct. Higher humidity indeed reduces evaporation. Our original statement should have referred to d-excess variations: a lower d-excess typically indicates higher humidity and weaker evaporation, while a higher d-excess reflects lower humidity and stronger evaporative fractionation (i.e., more intense non-equilibrium evaporation). We have revised the sentence as follows:

"Lower relative humidity in May and September resulted in higher d-excess values compared to other months, indicating more pronounced non-equilibrium evaporation during precipitation events."

315: The reference to Teo et al. 2021 seems to be missing from the reference list

**Response:** Thank you for the reminder. We will add this paper to the references (it was a preprint in 2021 and officially published in 2022).

**We have added the following reference to the manuscript:**

Teo, H. C., Raghavan, S. V., He, X., Zeng, Z., Cheng, Y., Luo, X., et al.: Large-scale reforestation can increase water yield and reduce drought risk for water-insecure regions in the Asia-Pacific, Glob. Change Biol., 28, 6385–6403, doi:10.1111/gcb.16404, 2022.

325: In this case, the groundwater amount decreases gradually with the T value increase.

Were any measurements performed of this decrease?

**Response:** Because collecting groundwater in high-altitude regions is quite challenging, this study **did not use any groundwater data**. Instead, we referenced findings from **other studies** conducted in areas similar to our own to support the theory presented in Figure 6. However, our current explanation is relatively brief and lacks reliability, so we have added the following details:

"Within the elevation range of 2543 to 3448 m, precipitation and seasonal permafrost are the main sources of groundwater recharge. Over the past 55 years, the seasonal permafrost has decreased at a rate of about 7.4 cm per decade (Li et al., 2016). Correlation analysis shows that when seasonal permafrost decreases, soil moisture in the upper layer increases (Qin et al., 2016). As vegetation transpiration increasingly consumes precipitation, soil water, and groundwater, the remaining groundwater becomes

more limited. Consequently, within this elevation range, the contribution of groundwater to runoff formation is minimal."

**In the reference list, all the journal names are missing**

**Response:**We have corrected the formatting of all references, with **specific details** as follows:

[revised manuscript text omitted]

**Responses to the reviewer's comments**

**Response to Reviewer #2**

Thank you very much for your recognition of the scientific issues, research value, and figures in this manuscript. How to better align the structure and content of the manuscript has been a persistent challenge for us, and your suggestions have helped us resolve this major issue. We fully agree with and appreciate your point about enhancing the clarity and readability of the manuscript through narrative and structure. After this revision, the quality of the manuscript will be greatly improved.

In order to enhance the quality of the manuscript, we have made every effort to carefully respond to each of your comments and made modifications in the manuscript. The red words, sentences, and subsections in the marked-up manuscript represent our editing changes.

**Main Comments:**

**Comment one:**

Clarity of the Storyline: I found the manuscript somewhat difficult to follow due to an unclear storyline. In particular, the structure of the introduction and discussion could be improved to better communicate the study's aims and main findings.

In the **introduction**, I only understood the objectives of the study in the final few sentences. Earlier parts — for example, lines 35–45 discussing moisture recycling — are interesting, but it's not clear how they tie into the rest of the manuscript. This section feels a bit out of balance and adds to the lack of clarity regarding the overall aim.

To improve this, I suggest organizing the introduction more clearly: start with the research context, then present the current state of knowledge and the knowledge gap, followed by the study's objectives. All necessary background information should be included, but try to **avoid** overwhelming the reader with too much detail at this early stage.

**Response:**As you mentioned, the introduction should better highlight the research topic, making the research background, objectives, and significance immediately clear. Therefore, we have reorganized the information that the introduction conveys to the readers. We made the following modifications for each paragraph of the introduction:

**First paragraph** (Research background: supplemented the principles related to stable isotopes while avoiding excessive details)

[revised manuscript text omitted]

**Supplementary references**

An Q, Liu L, Wang L, et al. Contribution of moisture recycling to water availability in China[J]. Water Resources Research, 2025, 61(4): e2024WR038054.

Cheng T F, Chen D, Wang B, et al. Human-induced warming accelerates local evapotranspiration and precipitation recycling over the Tibetan Plateau[J]. Communications Earth & Environment, 2024, 5(1): 388.

**Comment two:**

Discussion Section: The discussion could also benefit from a clearer structure. As it stands, it was difficult for me to identify the main findings of the research. The discussion appears to include new results rather than synthesizing and interpreting the key outcomes. The results already present quite a large amount of figures and tables. To me this was a bit overwhelming. I found it difficult to put each of the results in perspective. It would help if the authors began the discussion with a summary of the main findings, and then clearly relate those findings back to the study's results, objectives, and broader context.

Additionally, the manuscript touches on drought resilience and climate change in the introduction, but it's unclear how the results relate to these themes. Strengthening this connection in the discussion could significantly enhance the manuscript's impact.

**Response:** We understand the point you are trying to convey. The complex explanations and figures in the discussion section have burdened the understanding of the main research content, **particularly in subsection 5.1**. After careful consideration, we decided to move Table 5 to the supplementary material of the manuscript to help reduce confusion in this part.

Additionally, we focused on optimizing the discussion section from the following three aspects:

**PART 1** Summarizing the findings in the discussion section and integrating them with the main results.

**5.1.1 Contribution to recirculating water vapour in precipitation**

[revised manuscript text omitted]

**References**

Quan Q, He N, Zhang R, et al. Plant height as an indicator for alpine carbon sequestration and ecosystem response to warming[J]. Nature Plants, 2024, 10(6): 890-900.

**Comment three:**

Methodology Clarity: As a non-expert in isotope analysis, I found the methodological subsections clear in terms of explaining how individual variables were calculated — which I appreciated. However, I felt there was a lack of integration across these sections. Including a short overview at the beginning or end of the methodology section explaining how the different equations and steps fit together into the overall methodological framework would be very helpful. This could be especially valuable in Section 3.2, where a brief synthesis could guide the reader through the structure more effectively.

**Response:**We added a paragraph after 3.3 Research Methods that briefly describes the methods used, outlines the methodological framework, and explains the application of each method in the research. The supplementary details are as follows:

"First, determining the isotopic composition of water vapor formed from precipitation, vegetation, and

soil evaporation serves as the foundation for applying different models. Based on the isotopic values of different water vapor sources, the contribution of vegetation's evapotranspiration to the overall ecosystem evaporation can be established, which is a step in identifying the key factors affecting precipitation. Next, an end-member mixing model is used to quantify the contribution ratio of recycled water vapor in precipitation. The results of this analysis will be used to assess the impact of these key factors on the formation of precipitation. "

**Smaller comments:**

First, the abstract omits to say that this manuscript consults isotopic data. Could the authors please explain their methodology in a bit more detail here.

**Response:**This is reasonable and necessary. We have added information about the isotope methods and the isotopic data used in the abstract section. The details are as follows:

"The stable isotopes in water bodies play a crucial role in determining the evapotranspiration capacity of ecosystems and the mechanisms of precipitation formation. Between 2018 and 2022, we conducted research in the northeastern Qinghai-Tibet Plateau, collected and analyzed stable isotope water data from precipitation, soil water, and Picea crassifolia xylem water to quantify the impact of vegetation transpiration and recirculated water vapor on precipitation. "

Second, Could the authors please improve the readability of the figures. In some figures the text is quite small which makes it difficult to read them. In addition, could the authors please improve the captions of the figures and tables. I believe that if the authors include some more details here the readability of the figures would improve.

**Response:**We have reviewed every figure and table in the manuscript and made the following modifications:

[Figure]

**Figure 1:** Overview of the study area. (a) Geographical location of the study area, (b) Growth status of Qinghai spruce, and (c) Seasonal variation of meteorological conditions.

**Table 1:** Sampling location, the meteorological background, and sampling quantity information during the growing season.

[Figure]

**Figure 3:** (a) The average variation of d-excess during the growing season within the gradient of 2543 to 3448 m; (b), (c), and (d) represent the soil moisture content at different depths at elevations of 2543 m, 2721 m, and 3448 m, respectively.

**Table 3:** The isotopic composition of vegetation transpiration($\delta_T$), soil evaporation($\delta_E$), and ecosystem evapotranspiration($\delta_{ET}$) at different elevations during the growing season( * represents missing value).

**Figure 5:** (a), (b), (c), and (d) represent the proportion of soil evaporation and vegetation transpiration in the evapotranspiration of the ecosystem at different sampling points (0 represents a missing value).

Finally, for line 26-27, why is that? Could the authors please include a reference here?

**Response:** The expression here is indeed somewhat vague. After revisiting the supporting evidence from references— "Global water availability boosted by vegetation-driven changes in atmospheric moisture transport" and "The role of ecosystem transpiration in creating alternate moisture regimes by influencing atmospheric moisture convergence" — we found that the relationship between evapotranspiration and global water availability should be context-dependent. Accordingly, we have revised the sentence as follows:

"The impact of forest evapotranspiration on atmospheric moisture convergence for precipitation depends on ambient moisture conditions (Makarieva et al., 2023). At different spatial scales, vegetation transpiration capacity and local water availability exhibit varying relationships. Enhanced

evapotranspiration contributes up to 45% of available water in both local and downwind regions, although this ratio may differ in water-scarce areas (Cui et al., 2022)."

This sentence should be considered part of the **research status**; therefore, we have moved it to the **second paragraph**.

**Responses to the reviewer's comments**

**Response to Reviewer #1**

Thank you very much for your recognition of the revised manuscript. Your insights in both rounds of review have deepened our thinking on several key issues. Throughout the revision process, we learned to approach problems from different perspectives and endeavored to engage more comprehensively with your ideas, gaining many new insights. Your comments have been invaluable not only for improving this manuscript but also for enhancing our own way of thinking.

We will address each of your current comments in turn. Thank you again for acknowledging the scientific merits of the paper.

**Responses to the major comments**

**Comment one:**

-- I accept the authors' argument about local evaporation not influencing the isotopic ratio of downwind precipitation. As the cause, they quote other studies that state that evaporated moisture is ventilated out from the region without contributing to precipitation. As I understand, this could be due to the fact that evaporation is a slow process distributed in time, while precipitation events are more compact, so the amount of moisture evaporating during these short precipitation events should relatively small and may not impact the isotopic ratio significantly.

**Response:** This compelling perspective has greatly inspired me. **The fact that precipitation and evaporation occur at fundamentally different times is indeed the key reason why precipitation isotope ratios are not influenced by upwind evaporative moisture.**

**Comment two:**

-- I am still not very clear regarding the authors' explanations about the ultimate source of moisture for the forest if precipitation is consistently lower than evapotranspiration. The authors mention snowmelt, but snow is also part of precipitation. Do they mean that the forest is fed by the water streaming down the mountains? That is, that there is an inflow of liquid water into the area, which compensates for the mismatch between precipitation and evapotranspiration? Or maybe the authors' result is representative of the two years of measurements only, and there can be other years when precipitation exceeds evapotranspiration and soil moisture is replenished? Please consider adding a brief explanation on this point.

Note that in a steady state Runoff = Precipitation - Evapotranspiration, so if the latter sum is negative, this means that either soil moisture is declining or that there is an input of liquid water to the ecosystem (like irrigation or snowmelt streaming from the upper mountains).

**Response:** Thank you for your reminder—our previous explanation may indeed have been insufficiently clear. First, as you noted, precipitation can be divided into liquid and solid phases, and the samples we collected in the study area were entirely liquid. Second, we agree with your point on water balance:

when Precipitation minus Evapotranspiration (P – ET) is negative in the study area, liquid snowmelt from upstream flows in to restore the water budget. This influx is the key reason why the forest can survive here. We will add a brief clarification in Section 5.1.2, **as follows**:

"However, to support forest growth and maintain the regional water balance, upstream snowmelt compensates for the deficit between precipitation and evapotranspiration and flows into the forest ecosystem as a supplemental water source."

**Responses to the minor comments**

-- Please check through the text as there are many cases when words stick together with braces without spaces, like "liquid phase(Dansgaard et al., 1964)", "expression(Kool et al", "follow(Kong et al., 2013" (should be follows).

**Response:** We have reviewed the entire manuscript and corrected these errors.

-- Please mention in the text that Picea crassifolia is the Qinghai spruce (as indicated in the legend to Fig. 1b).

**Response:** After reviewing the relevant literature, we found that "Qinghai spruce" is not a standardized botanical name. Accordingly, we have replaced every instance of it in the manuscript with the scientific name "**Picea crassifolia**."